# Mitigating Label Shift in Tabular In-Context Learning via Test-Time Posterior Adjustment

Seunghan Lee [1]

## Abstract

TabPFN has recently gained attention as a foundation model for tabular datasets, achieving strong performance by leveraging in-context learning on synthetic data. However, we find that TabPFN is vulnerable to *label shift*, often overfitting to the majority class in the training dataset. To address this limitation, we propose *DistPFN*, the first test-time posterior adjustment method designed for tabular foundation models. DistPFN rescales predicted class probabilities by downweighting the influence of the training prior (i.e., the class distribution of the context) and emphasizing the contribution of the model's predicted posterior, without architectural modification or additional training. We further introduce *DistPFN-T*, which incorporates temperature scaling to adaptively control the adjustment strength based on the discrepancy between prior and posterior. We evaluate our methods on over 250 OpenML datasets, demonstrating substantial improvements for various TabPFN-based models in classification tasks under label shift, while maintaining strong performance in standard settings without label shift. Code is available at this repository: `https://github.com/seunghan96/DistPFN`.

## 1. Introduction

Tabular data is among the most prevalent data formats across various domains, such as healthcare (Johnson et al., 2016) and finance (Arun et al., 2016). Tree-based models (Chen & Guestrin, 2016; Ke et al., 2017) have consistently demonstrated strong performance on tabular tasks, owing to their ability to handle heterogeneous feature types with minimal hyperparameter tuning. Recently, deep learning (DL) methods, especially transformer-based models (Huang et al.,

[1]LG AI Research, Seoul, South Korea. Correspondence to: Seunghan Lee <seunghan.lee@lgresearch.ai>.

*Proceedings of the 43rd International Conference on Machine Learning*, Seoul, South Korea. PMLR 306, 2026. Copyright 2026 by the author(s).

*Table 1.* **Confusion matrices.** TabPFN exhibits severe *majority-class bias*, predicting **98.3%** of samples as the **majority class**, while DistPFN mitigates this via a simple test-time adjustment.

| TabPFN-v2 (Nature 2025) | | | + DistPFN (Ours) | | |
|---|---|---|---|---|---|
| | $\hat{y} = 0$ | $\hat{y} = 1$ | (%) | $\hat{y} = 0$ | $\hat{y} = 1$ |
| $y = 0$ | **87.2%** | 0.0% | $y = 0$ | 81.1% | 6.1% |
| $y = 1$ | **11.1%** | 1.7% | $y = 1$ | 3.0% | 9.8% |
| Total | **98.3%** | 1.7% | Total | 84.1% | 15.9% |

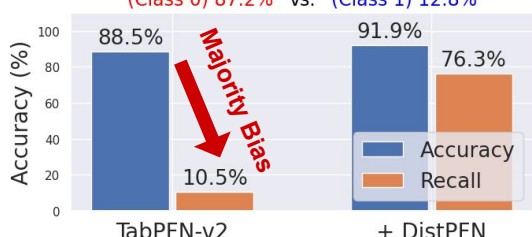

*Figure 1.* **Majority-class bias.** TabPFN suffers from majority-class bias, resulting in poor recall for the minority class.

2020; Gorishniy et al., 2021), have emerged as strong alternatives by capturing complex feature interactions.

Among these methods, TabPFN (Hollmann et al., 2023) introduces in-context learning (ICL) to tabular classification by pretraining on synthetic datasets and producing predictions for test samples in a single forward pass. While TabPFN achieves strong performance on small-scale datasets, it suffers from scalability issues due to the quadratic complexity of self-attention (Vaswani et al., 2017). To address this limitation, several extensions have been proposed to improve inference efficiency on larger datasets (Thomas et al., 2024; Xu et al., 2025; Zeng et al., 2025).

In this paper, we highlight an overlooked limitation of TabPFN, namely its vulnerability to *label shift*, which is a critical scenario in tabular learning (kim). We observe that TabPFN tends to overfit to the majority class in the training dataset (i.e., *majority-class bias*), resulting in poor performance when the class distribution in the test dataset differs. As shown in Table 1 and Figure 1, TabPFN-v2 (Hollmann

*Table 2.* **Applicability of DistPFN.** DistPFN is designed for models that *explicitly* reference training data (e.g., tabular FMs), while it is technically applicable to any classifier that produces probabilistic outputs, including tree-based models.

| Prior | Encoded In | Representative Methods |
|---|---|---|
| Explicit | Training dataset $p_\theta(x, \mathcal{D}_{\text{train}})$ | TabPFN, TabPFN-v2, LoCalPFN, TabICL, $k$NN |
| Implicit | Model parameters $p_\theta(x)$ | Random Forest, XGBoost, MLP, Logistic Regression |

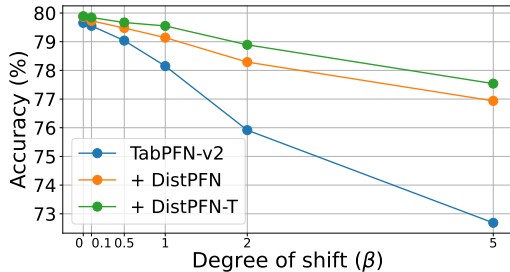

*Figure 3.* Robustness to shift with DistPFN.

**(A)** 1) Train → 2) Test | **(B)** (Zero-shot) Test

Classical models → Classical models | TabPFN-based models

Train — Majority / Minority / Test

(A) Refer to train-set **implicitly** (via model params)
(B) Refer to train-set **explicitly** (via attention)
→ Can utilize the **statistics of train-set** directly!

*Figure 2.* Direct utilization of train-set in TabPFN-based models.

et al., 2025) exhibits a majority-class bias, making incorrect predictions even when trained and tested on the *same* dataset (CostaMadre1 (Bischl et al., 2017)).

To this end, we propose *DistPFN*, a simple yet effective test-time adaptation method that improves the robustness of TabPFN-based models to label shift, without modifying the architecture or updating any parameters. Specifically, it adjusts the model's output distribution (i.e., **posterior**) by reweighting class probabilities based on the ratio between the posterior and the class distribution of the training dataset (i.e., **prior**). Intuitively, this adjustment downweights the influence of the training distribution and amplifies the impact of the observed test samples. Furthermore, to make the adjustment more adaptive to distributional mismatch between the labels of the training and test datasets, we propose *DistPFN-T*, which adjusts the reweighting intensity via temperature scaling, where the temperature is determined based on the discrepancy between the posterior and the prior.

Unlike prior correction methods that require estimating test priors, our approach is *specifically designed for tabular FMs* rather than classical models, where the training prior is explicitly encoded through in context inference as shown in Table 2 and Figure 2. By leveraging this explicit prior usage, our adjustment operates directly on the model's posterior, without additional training, external prior estimation, or architectural changes, making it naturally aligned with the inference mechanism of tabular FMs.

We conduct extensive experiments on over 250 classifica-

tion datasets to evaluate the effectiveness of DistPFN in both standard and label-shifted classification scenarios. As shown in Figure 3, DistPFN significantly improves accuracy of TabPFN-v2 (Hollmann et al., 2025) under label shift, with the x-axis indicating the degree of shift (see Sec. 4.4) and the y-axis showing average accuracy over 253 datasets. Our main contributions are summarized as follows:

- We identify an overlooked limitation of TabPFN under *label shift*, where it becomes biased toward the majority class in the training dataset, leading to severe performance degradation as the shift increases.
- We propose **DistPFN**, a novel and simple test-time adaptation method that adjusts TabPFN's output distribution based on the ratio between the posterior and the prior. We further introduce **DistPFN-T**, which extends this approach by applying temperature scaling to adaptively control the strength of adjustment according to the distributional mismatch between the prior and the posterior.
- We present extensive evaluations on over 250 classification datasets, demonstrating that our methods significantly improve the performance of various TabPFN-based models under label shift, surpassing baseline methods and achieving state-of-the-art (SoTA) performance.
- We provide a theoretical interpretation of our method, showing that it can be viewed from both 1) classical label shift correction and 2) Bayesian inference, with details provided in Appendix D.

## 2. Related Works

**Gradient Boosting Decision Trees (GBDTs).** Gradient Boosting Decision Trees (GBDTs), including XGBoost (Chen & Guestrin, 2016), LightGBM (Ke et al., 2017), and CatBoost (Prokhorenkova et al., 2018), are widely used for tabular data due to their strong inductive biases, minimal preprocessing, and robust performance across diverse datasets (Grinsztajn et al., 2022; McElfresh et al., 2023). Despite advances in deep learning, GBDTs remain dominant in tabular tasks due to the difficulty of encoding inductive biases through gradient-based training, often leading to inferior performance in benchmarks (Grinsztajn et al., 2022; Shwartz-Ziv & Armon, 2022; McElfresh et al., 2023).

**Tabular Deep Learning (DL).** Transformer-based models have been proposed to capture complex feature interactions in tabular data, where TabTransformer (Huang et al., 2020) applies contextual embeddings to categorical features using self-attention. Several other architectures have also been introduced (Arik & Pfister, 2021; Somepalli et al., 2021), but these methods typically require extensive hyperparameter tuning and often fail to generalize across datasets (Kadra et al., 2021; Grinsztajn et al., 2022). Recently, TabM (Gorishniy et al., 2024a) proposes an MLP-based ensemble model that generates multiple predictions per instance with shared parameters, and ModernNCA (Ye et al., 2024) introduces a differentiable $k$NN approach based on neighborhood components analysis. RealMLP (Holzmüller et al., 2024) simplifies MLPs with improved design and meta-tuned default parameters, and TabR (Gorishniy et al., 2024b) augments inputs by retrieving similar training examples.

**Tabular Foundation Models (FMs).** TabPFN (Hollmann et al., 2023) is a foundation model for tabular classification that uses in-context learning (ICL) via synthetic pretraining. It predicts test instances by conditioning on training inputs and labels, without gradient updates. TabPFN-v2 (Hollmann et al., 2025) enhances scalability and generalization through dual-axis attention over samples and features. Limited by its computational complexity, TabPFN has led to several extensions, where LoCalPFN (Thomas et al., 2024) improves TabPFN by retrieving neighbors of test samples and fine-tuning on this local context, and MixturePFN (Xu et al., 2025) scales TabPFN to larger datasets by combining nearest-neighbor sampling with bootstrapped fine-tuning. TuneTables (Feuer et al., 2024) scales TabPFN to larger datasets by learning dataset-specific contexts through fine-tuning, and TabFlex (Zeng et al., 2025) replaces softmax attention of TabPFN with linear attention to improve efficiency. TabICL (Qu et al., 2025) uses a two-stage architecture with column-then-row attention to embed rows.

**Label shift.** Several studies have addressed label shift by rescaling classifier outputs, typically requiring estimation of the test distribution (Elkan, 2001; Lipton et al., 2018; Azizzadenesheli et al., 2019). In contrast, our method avoids test prior estimation and instead leverages only the training prior and the predicted distribution, yielding a simple and efficient plug-in adjustment that requires no architectural modifications. Moreover, while Drift-Resilient TabPFN (Helli et al., 2024) addresses *temporal* shift through *pretraining*, our method addresses *label* shift and enables test-time adaptation of pretrained models *without retraining*. Specifically, compared to classical post-hoc methods such as EME (Saerens et al., 2002), which requires iterative EM estimation of the test prior, DistPFN requires no test prior estimation and operates in a single forward pass. Unlike training-time methods such as logit adjustment (Menon et al., 2021) and balanced softmax (Ren et al.,

*Table 3.* **vs. Label shift methods.** DistPFN operates in a post-hoc manner without test prior estimation or model retraining.

|  | EME | Logit adj. | Bal. softmax | **DistPFN** |
|---|---|---|---|---|
| *When applied* | Inference | Training | Training | **Inference** |
| *Test prior estimation* | ✓ | ✗ | ✗ | ✗ |
| *Model retraining* | ✗ | ✓ | ✓ | ✗ |

2020), DistPFN requires no model retraining, making it applicable to any pretrained tabular FM as a plug-in module. A comparison of label shift methods are shown in Table 3.

## 3. Preliminaries

**Tabular classification.** A tabular dataset consists of instances $x_i \in \mathbb{R}^d$, where each $x_i$ is a $d$-dimensional feature vector composed of numerical or categorical attributes. Each instance is associated with a label $y_i \in \{1, \ldots, C\}$ indicating one of $C$ predefined classes. The training dataset is denoted as $\mathcal{D}_{\text{train}} = \{(x_i, y_i)\}_{i=1}^{N_{\text{train}}}$, and the test dataset as $\mathcal{D}_{\text{test}} = \{x_j\}_{j=1}^{N_{\text{test}}}$. In a tabular classification task, a model $f$ predicts the class labels $y_j$ given the $x_j$ from the test set.

**Tabular ICL.** TabPFN-based methods (Hollmann et al., 2023; 2025) predict test labels by conditioning on the labeled $\mathcal{D}_{\text{train}} = \{(x_i, y_i)\}_{i=1}^{N_{\text{train}}}$ and the unlabeled $\mathcal{D}_{\text{test}} = \{x_j\}_{j=1}^{N_{\text{test}}}$. Note that these models infer the test labels $y_j$ in a single forward pass without gradient updates.

**Label shift.** In this paper, we aim to address label shift, which is a distribution shift scenario where the marginal label distribution differs between training and test datasets, i.e., $p_{\text{train}}(y) \neq p_{\text{test}}(y)$. Specifically, for each class $c_k \in \{1, \ldots, C\}$, the label distributions are estimated as

$$p_{\text{train/test}}(y = c_k) = \frac{|\{i \in [N_{\text{train/test}}] : y_i = c_k\}|}{N_{\text{train/test}}}, \quad (1)$$

where $p_{\text{test}}(y = c_k)$ cannot be directly computed, as test labels are not observed.

## 4. Methodology

In this section, we build upon **TabPFN** [1] (Sec. 4.1), which performs ICL by conditioning on the training dataset but suffers from label shift. To address this, we propose **DistPFN** (Sec. 4.2), which reweights predicted class probabilities using the ratio between the posterior and the prior. We further introduce **DistPFN-T** (Sec. 4.3), which adaptively controls the adjustment strength using temperature scaling. Lastly, we propose a **inverse-frequency-based oversampling** method (Sec. 4.4) that oversamples rare classes in the training dataset based on inverse frequency to enable controlled evaluation under label shift. The overall framework of our method is described in Figure 4.

---

[1]We use **TabPFN** to refer broadly to the *family of ICL-based tabular FMs* (Hollmann et al., 2025; Qu et al., 2025).

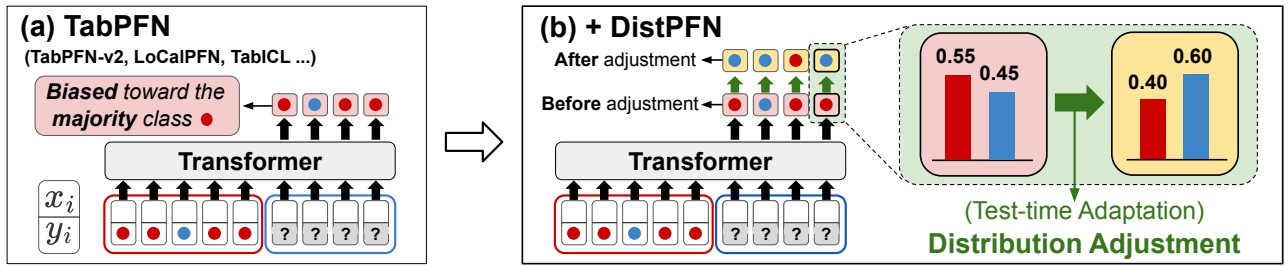

*Figure 4.* **Overall framework of DistPFN.** (a) **TabPFN** exhibits a *majority-class bias* under label shift, predicting test instances toward the majority class in the training dataset. (b) **DistPFN** mitigates this bias via a simple *test-time adaptation* method that rescales the predicted class probabilities for each test instance.

## 4.1. TabPFN

In tabular in-context learning, TabPFN predicts the label of a test instance $x_j$ by conditioning on the entire training dataset $\mathcal{D}_{\text{train}}$. Let $f(x_j, \mathcal{D}_{\text{train}}) \in \mathbb{R}^C$ denote the model output logits for the $C$ classes. The posterior distribution of TabPFN is computed via softmax:

$$\widehat{p}_{\text{TabPFN}}(y \mid x_j, \mathcal{D}_{\text{train}}) = \frac{\exp\left(f(x_j, \mathcal{D}_{\text{train}})[y]\right)}{\sum_{c=1}^{C} \exp\left(f(x_j, \mathcal{D}_{\text{train}})[c]\right)}, \quad (2)$$

where $[\cdot]$ denotes indexing over class logits. For simplicity, we denote the posterior of any model $\widehat{p}(y \mid x_j, \mathcal{D}_{\text{train}})$ as $\widehat{p}(y)$, omitting the notations of input and training dataset.

## 4.2. DistPFN: Test-Time Posterior Adjustment

To improve the robustness under label shift, DistPFN extends TabPFN by adjusting the model's output, or the predicted distribution of $x_j$. Specifically, it introduces an adjustment factor based on the ratio between the predicted distribution $\widehat{p}_{\text{TabPFN}}(y)$ and the training prior $p_{\text{train}}(y)$ as:

$$\widetilde{p}_{\text{DistPFN}}(y) = \text{Norm}(\widehat{p}_{\text{TabPFN}}(y) \cdot \underbrace{\frac{\widehat{p}_{\text{TabPFN}}(y)}{p_{\text{train}}(y)}}_{\text{Adjustment factor } (\alpha)}) \quad (3)$$

$$= \text{Norm}(\frac{\widehat{p}_{\text{TabPFN}}(y)^2}{p_{\text{train}}(y)}), \quad (4)$$

where $\text{Norm}(\cdot)$ denotes normalization over classes to ensure the probabilities sum to one. This adjustment down-weights the influence of the training prior and instead emphasizes the model's own prediction at test time, without requiring any modification to the model architecture or parameters. We note that this constitutes a *partial* correction rather than a full one. As the predicted posterior $\widehat{p}(y)$ does not fully collapse to the prior, dividing by $p_{\text{train}}(y)$ reduces the influence of the prior only to a limited extent.

## 4.3. DistPFN-T: Temperature-Scaled Adjustment

While DistPFN corrects for label shift using the training prior, the optimal strength of adjustment may vary depending on *the deviation of test-time predictions from the training*

*Table 4.* Prediction w/ DistPFN and DistPFN-T.

| Prediction | Case 1) **Majority** | | Case 2) **Minority** | |
|---|---|---|---|---|
| Class (Prior) | A (0.8) | B (0.2) | A (0.8) | B (0.2) |
| TabPFN | **0.60** | 0.40 (-) | 0.40 | **0.60** (-) |
| + DistPFN | 0.36 | **0.64** (↑) | 0.10 | **0.90** (↑↑) |
| + DistPFN-T | 0.33 | **0.67** (↑↑) | 0.12 | **0.88** (↑) |

*prior.* To make posterior adjustment more adaptive to the discrepancy between the training and the test dataset, we propose DistPFN-T, which introduces temperature scaling to control the sharpness of adjustment based on the discrepancy between the training prior $p_{\text{train}}(y)$ and the predicted distribution $\widehat{p}_{\text{TabPFN}}(y)$. Specifically, a temperature value $\tau$ is computed using the cross-entropy (CE) between the two distributions, where the predicted distribution $\widehat{p}_{\text{TabPFN}}(y)$ is then passed through a temperature-scaled softmax as:

$$\widehat{p}_{\text{TabPFN-T}}(y = c) = \frac{\exp\left(\widehat{p}_{\text{TabPFN}}(y = c)/\tau\right)}{\sum_{c'=1}^{C} \exp\left(\widehat{p}_{\text{TabPFN}}(y = c')/\tau\right)}, \quad (5)$$

where $\tau = \text{CE}(\widehat{p}_{\text{TabPFN}}(y), p_{\text{train}}(y))$.

When the predicted distribution strongly deviates from the training prior (i.e., high cross-entropy), a high temperature is applied to smooth the predictions, thereby preventing over-adjustment. This temperature-scaled distribution $\widehat{p}_{\text{TabPFN-T}}(y)$ is used as the numerator of the adjustment factor in DistPFN-T, replacing $\widehat{p}_{\text{TabPFN}}(y)$ used in DistPFN as:

$$\widetilde{p}_{\text{DistPFN-T}}(y) = \text{Norm}(\widehat{p}_{\text{TabPFN}}(y) \cdot \underbrace{\frac{\widehat{p}_{\text{TabPFN-T}}(y)}{p_{\text{train}}(y)}}_{\text{Adjustment factor } (\alpha)}). \quad (6)$$

Table 4 shows the example where DistPFN-T smooths the adjusted probabilities of DistPFN based on the prediction of TabPFN and the training prior. When TabPFN's prediction aligns with the majority class, as in **Case 1) Majority**, DistPFN-T amplifies the *minority* class more strongly than DistPFN, whereas when it aligns with the minority class, as in **Case 2) Minority**, DistPFN-T amplifies the *majority* prediction to mitigate overprediction. This design is intuitive, as it counterbalances the bias introduced by the label

**Algorithm 1** Pseudocode for DistPFN

```
# x_test: test instance(s)
# D_train: training dataset
# p_train: training class prior
# f: TabPFN-based model
# alpha: adjustment factor

logits = f(x_test, D_train)
p_hat = softmax(logits, dim=0)

if method == "tabpfn":
    alpha = 1

elif method == "distpfn":
    alpha = p_hat / p_train

elif method == "distpfn-t":
    tau = cross_entropy(p_hat, p_train)
    p_hat_scaled = softmax(p_hat / tau, dim=0)
    alpha = p_hat_scaled / p_train

p_hat = normalize(alpha * p_hat)
```

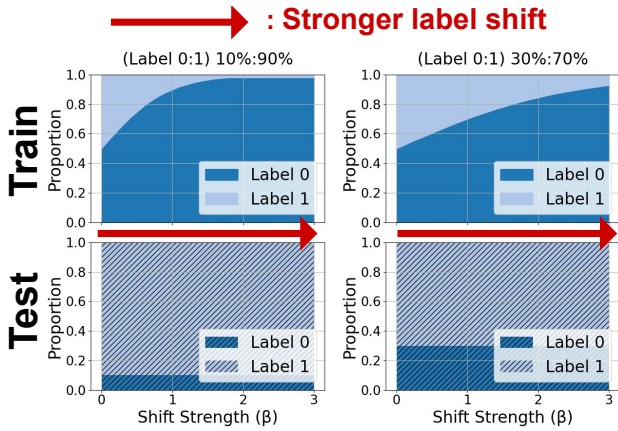

Figure 5. **Inverse-frequency-based oversampling.** As $\beta$ increases, the training distribution becomes increasingly biased toward rare classes, while the test distribution remains fixed

distribution and the model's own prediction, leading to more calibrated and stable outputs. The effectiveness of DistPFN-T over DistPFN is demonstrated in Table 5 and Figure 7, using three different FMs under varying degrees of shift.

As TabPFN takes the entire test dataset (i.e., multiple instances) as input at once rather than processing each instance individually (i.e., single instance), the proposed methods apply their adjustment based on either the predicted distribution of a single instance or the average distribution of the test dataset, with the latter used by default in our experiments. A comparison of these two strategies is presented in Table 9, demonstrating the robustness of this design choice.

### 4.4. Benchmark for Label Shift

To enable controlled evaluation under label shift, we propose *inverse-frequency-based oversampling*, which modifies the label distribution of $\mathcal{D}_{\text{train}}$ by oversampling each class based on its inverse frequency, while keeping $\mathcal{D}_{\text{test}}$ unchanged for fair comparison with the non-shifted setting. Note that we adopt oversampling rather than undersampling to avoid potential performance degradation caused by the removal of training instances. Specifically, we assign assigning higher weights to rarer classes in the original distribution as:

$$w_k = \left( \frac{1}{p(y = c_k)} \right)^{\beta}, \qquad \tilde{w}_k = \frac{w_k}{\sum_{j=1}^{C} w_j},$$

with $\beta \geq 0$ controlling the strength of the shift.

Figure 5 illustrates the label distributions of the training/test datasets after oversampling, with respect to the shift strength ($\beta$) and the class distribution of the entire dataset. Higher $\beta$ assigns higher sampling probabilities to rare classes, inducing a stronger shift, whereas $\beta = 0$ corresponds to uniform sampling. Note that $\beta = 0$ does not imply the absence of label shift, as the dataset itself may be class-imbalanced.

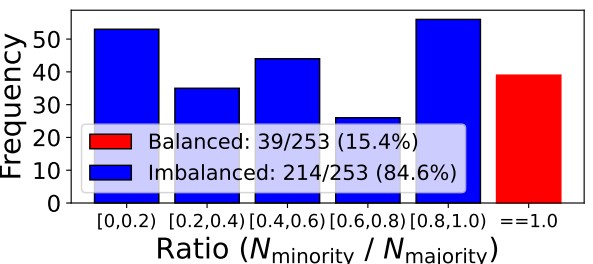

Figure 6. Class balance ratio of 253 OpenML datasets.

## 5. Experiments

### 5.1. Experimental Setup

**Task and metrics.** We evaluate our methods on tabular classification tasks both with and without label shift. For the evaluation metrics, we employ accuracy (Acc.) and average rank (Rank), following the previous works (Hollmann et al., 2023). We additionally report expected calibration error (ECE) and precision to provide evaluation tailored to class imbalance. Further details are provided in Appendix A.

**Datasets.** We evaluate our methods on 253 tabular classification datasets from OpenML (Bischl et al., 2017), which span a wide range of feature dimensions, class cardinalities, sample sizes, and domains. Unless otherwise specified, performance is reported as the mean accuracy across all datasets, averaged over five random seeds. Additionally, to specifically assess performance under label shift, we construct synthetic variants by modifying the test set while keeping the training set fixed, for fair comparison with standard setup, as described in Section 4.4. Following the previous works (Hollmann et al., 2023; 2025), all methods are evaluated using the fixed train/test splits, where each dataset is split into 50% training and 50% test data.

*Table 5.* **Tabular classification results.** While most baselines suffer substantial performance degradation under label shift, our methods significantly improve the accuracy of tabular FMs (e.g., TabPFN-v2) across varying degrees of shift ($\beta$), averaged over 253 datasets.

| | | Methods | w/o shift | Shift strength ($\beta$) | | | | | | Avg. |
|---|---|---|---|---|---|---|---|---|---|---|
| | | | | 0.0 | 0.1 | 0.5 | 1.0 | 2.0 | 5.0 | |
| Machine Learning | | LogReg. | $0.765_{\pm0.002}$ | $0.719_{\pm0.002}$ | $0.709_{\pm0.002}$ | $0.674_{\pm0.004}$ | $0.634_{\pm0.004}$ | $0.597_{\pm0.002}$ | $0.566_{\pm0.004}$ | 0.650 |
| | | + HPO | $0.771_{\pm0.003}$ | $0.697_{\pm0.005}$ | $0.687_{\pm0.003}$ | $0.653_{\pm0.003}$ | $0.616_{\pm0.006}$ | $0.586_{\pm0.003}$ | $0.550_{\pm0.003}$ | 0.631 |
| | | SVM | $0.780_{\pm0.003}$ | $0.684_{\pm0.002}$ | $0.646_{\pm0.004}$ | $0.560_{\pm0.005}$ | $0.531_{\pm0.003}$ | $0.486_{\pm0.004}$ | $0.448_{\pm0.004}$ | 0.559 |
| | | + HPO | $0.784_{\pm0.003}$ | $0.731_{\pm0.001}$ | $0.689_{\pm0.008}$ | $0.626_{\pm0.003}$ | $0.597_{\pm0.005}$ | $0.570_{\pm0.005}$ | $0.541_{\pm0.007}$ | 0.626 |
| | | MLP | $0.778_{\pm0.004}$ | $0.658_{\pm0.005}$ | $0.647_{\pm0.004}$ | $0.613_{\pm0.005}$ | $0.574_{\pm0.006}$ | $0.527_{\pm0.004}$ | $0.493_{\pm0.001}$ | 0.585 |
| | | + HPO | $0.795_{\pm0.005}$ | $0.706_{\pm0.006}$ | $0.688_{\pm0.006}$ | $0.654_{\pm0.008}$ | $0.615_{\pm0.006}$ | $0.580_{\pm0.005}$ | $0.545_{\pm0.005}$ | 0.631 |
| | | $k$NN | $0.765_{\pm0.004}$ | $0.663_{\pm0.004}$ | $0.657_{\pm0.004}$ | $0.629_{\pm0.003}$ | $0.589_{\pm0.003}$ | $0.538_{\pm0.003}$ | $0.501_{\pm0.004}$ | 0.596 |
| | | + HPO | $0.783_{\pm0.002}$ | $0.693_{\pm0.002}$ | $0.684_{\pm0.003}$ | $0.644_{\pm0.004}$ | $0.588_{\pm0.003}$ | $0.540_{\pm0.003}$ | $0.498_{\pm0.002}$ | 0.608 |
| | | Random Forest | $0.796_{\pm0.003}$ | $0.768_{\pm0.003}$ | $0.765_{\pm0.003}$ | $0.748_{\pm0.005}$ | $0.718_{\pm0.004}$ | $0.665_{\pm0.005}$ | $0.618_{\pm0.005}$ | 0.714 |
| | | + HPO | $0.803_{\pm0.002}$ | $0.771_{\pm0.002}$ | $0.767_{\pm0.001}$ | $0.743_{\pm0.004}$ | $0.701_{\pm0.004}$ | $0.627_{\pm0.008}$ | $0.578_{\pm0.006}$ | 0.698 |
| | | LightGBM | $0.789_{\pm0.003}$ | $0.758_{\pm0.004}$ | $0.753_{\pm0.002}$ | $0.734_{\pm0.003}$ | $0.705_{\pm0.004}$ | $0.657_{\pm0.005}$ | $0.618_{\pm0.005}$ | 0.704 |
| | | + HPO | $0.790_{\pm0.006}$ | $0.726_{\pm0.008}$ | $0.661_{\pm0.005}$ | $0.655_{\pm0.008}$ | $0.608_{\pm0.008}$ | $0.577_{\pm0.015}$ | $0.551_{\pm0.004}$ | 0.630 |
| | | CatBoost | $0.803_{\pm0.001}$ | $0.774_{\pm0.002}$ | $0.771_{\pm0.002}$ | $0.751_{\pm0.004}$ | $0.718_{\pm0.004}$ | $0.665_{\pm0.005}$ | $0.621_{\pm0.005}$ | 0.717 |
| | | + HPO | $0.802_{\pm0.002}$ | $0.774_{\pm0.002}$ | $0.771_{\pm0.002}$ | $0.752_{\pm0.004}$ | $0.719_{\pm0.004}$ | $0.665_{\pm0.006}$ | $0.621_{\pm0.005}$ | 0.717 |
| Deep Learning | Non-FMs | FT-Transformer | $0.784_{\pm0.002}$ | $0.748_{\pm0.004}$ | $0.746_{\pm0.004}$ | $0.718_{\pm0.005}$ | $0.674_{\pm0.005}$ | $0.610_{\pm0.003}$ | $0.551_{\pm0.007}$ | 0.675 |
| | | TabM | $0.794_{\pm0.002}$ | $0.762_{\pm0.004}$ | $0.757_{\pm0.004}$ | $0.735_{\pm0.003}$ | $0.694_{\pm0.005}$ | $0.624_{\pm0.006}$ | $0.565_{\pm0.006}$ | 0.690 |
| | | TabulaRNN | $0.749_{\pm0.003}$ | $0.699_{\pm0.003}$ | $0.684_{\pm0.003}$ | $0.641_{\pm0.004}$ | $0.585_{\pm0.009}$ | $0.522_{\pm0.011}$ | $0.465_{\pm0.008}$ | 0.599 |
| | | MambaTab | $0.719_{\pm0.004}$ | $0.629_{\pm0.006}$ | $0.603_{\pm0.004}$ | $0.525_{\pm0.002}$ | $0.466_{\pm0.010}$ | $0.430_{\pm0.005}$ | $0.394_{\pm0.002}$ | 0.508 |
| | | RealMLP | $0.794_{\pm0.002}$ | $0.760_{\pm0.004}$ | $0.758_{\pm0.005}$ | $0.745_{\pm0.003}$ | $0.720_{\pm0.005}$ | $0.677_{\pm0.002}$ | $0.643_{\pm0.004}$ | 0.717 |
| | FMs | LoCalPFN | $\mathbf{0.816}_{\pm0.002}$ | $0.794_{\pm0.003}$ | $0.793_{\pm0.004}$ | $0.788_{\pm0.003}$ | $0.778_{\pm0.002}$ | $0.753_{\pm0.004}$ | $0.719_{\pm0.000}$ | 0.771 |
| | | + DistPFN | $\mathbf{0.816}_{\pm0.002}$ | $\underline{0.797}_{\pm0.001}$ | $\underline{0.796}_{\pm0.002}$ | $\underline{0.794}_{\pm0.002}$ | $\underline{0.790}_{\pm0.002}$ | $\underline{0.782}_{\pm0.001}$ | $\underline{0.770}_{\pm0.003}$ | $\underline{0.788}$ |
| | | + DistPFN-T | $\mathbf{0.816}_{\pm0.002}$ | $\mathbf{0.798}_{\pm0.002}$ | $\mathbf{0.797}_{\pm0.002}$ | $\mathbf{0.796}_{\pm0.002}$ | $\mathbf{0.794}_{\pm0.002}$ | $\mathbf{0.787}_{\pm0.001}$ | $\mathbf{0.776}_{\pm0.003}$ | $\mathbf{0.791}$ |
| | | TabICL | $\mathbf{0.806}_{\pm0.002}$ | $0.783_{\pm0.003}$ | $0.781_{\pm0.003}$ | $0.770_{\pm0.003}$ | $0.747_{\pm0.003}$ | $0.704_{\pm0.006}$ | $0.664_{\pm0.006}$ | 0.742 |
| | | + DistPFN | $\mathbf{0.806}_{\pm0.002}$ | $\underline{0.786}_{\pm0.002}$ | $\underline{0.786}_{\pm0.002}$ | $\underline{0.781}_{\pm0.002}$ | $\underline{0.776}_{\pm0.002}$ | $\underline{0.763}_{\pm0.002}$ | $\underline{0.746}_{\pm0.004}$ | $\underline{0.773}$ |
| | | + DistPFN-T | $\mathbf{0.806}_{\pm0.003}$ | $\mathbf{0.786}_{\pm0.003}$ | $\mathbf{0.786}_{\pm0.003}$ | $\mathbf{0.783}_{\pm0.002}$ | $\mathbf{0.780}_{\pm0.002}$ | $\mathbf{0.771}_{\pm0.001}$ | $\mathbf{0.755}_{\pm0.004}$ | $\mathbf{0.777}$ |
| | | TabPFN-v2 | $\mathbf{0.818}_{\pm0.004}$ | $\underline{0.797}_{\pm0.003}$ | $0.796_{\pm0.004}$ | $0.790_{\pm0.002}$ | $0.782_{\pm0.002}$ | $0.759_{\pm0.003}$ | $0.727_{\pm0.003}$ | 0.775 |
| | | + DistPFN | $\mathbf{0.818}_{\pm0.002}$ | $\mathbf{0.799}_{\pm0.001}$ | $\underline{0.797}_{\pm0.002}$ | $\underline{0.795}_{\pm0.002}$ | $\underline{0.791}_{\pm0.003}$ | $\underline{0.783}_{\pm0.003}$ | $\underline{0.769}_{\pm0.003}$ | $\underline{0.789}$ |
| | | + DistPFN-T | $\mathbf{0.818}_{\pm0.002}$ | $\mathbf{0.799}_{\pm0.003}$ | $\mathbf{0.798}_{\pm0.002}$ | $\mathbf{0.797}_{\pm0.002}$ | $\mathbf{0.796}_{\pm0.003}$ | $\mathbf{0.789}_{\pm0.003}$ | $\mathbf{0.775}_{\pm0.003}$ | $\mathbf{0.792}$ |

*Figure 7.* Performance by shift.

*Figure 8.* Rank under shift ($\beta = 2$).

*Table 6.* Performance on TableShift.

| Label ratio 0:1 (%) | | Diabetes | Acsincome | Acspubcov |
|---|---|---|---|---|
| Train dataset | | 57.6:42.4 | 63.0:37.0 | 72.7:27.3 |
| Test dataset-IID | | 58.4:41.6 | 63.0:37.0 | 72.7:27.3 |
| Test dataset-OOD | | 50.6:49.4 | 59.1:40.9 | 63.9:36.1 |
| TabPFN-v2 | IID | 0.646 | 0.770 | 0.794 |
| | OOD | 0.589 | 0.795 | 0.699 |
| + DistPFN | IID | $\underline{0.648}$ | $\underline{0.773}$ | $\underline{0.797}$ |
| | OOD | $\underline{0.598}$ | $\underline{0.797}$ | $\underline{0.705}$ |
| + DistPFN-T | IID | $\mathbf{0.649}$ | $\mathbf{0.775}$ | $\mathbf{0.804}$ |
| | OOD | $\mathbf{0.600}$ | $\mathbf{0.799}$ | $\mathbf{0.706}$ |

**Class-imbalanced Benchmark.** We examine the degree of class imbalance across 253 OpenML datasets by defining the *balance ratio* as the number of samples in the minority class ($N_{\text{minority}}$) divided by the number of samples in the majority class ($N_{\text{majority}}$). A balance ratio of 100% corresponds to a perfectly balanced dataset, while lower values indicate increasing imbalance. As shown in Figure 6, approximately 85% of the datasets exhibit class imbalance, highlighting the importance of addressing the majority-class bias.

### 5.2. Tabular Classification

Table 5 reports the average accuracy over 253 datasets across six levels of label shift ($\beta$), comparing our method against 16 baselines, including three tabular FMs to which our method

is applied. For LoCalPFN, we use 10 nearest neighbors for each test sample. While maintaining the original performance under the standard setting without label shift, our method substantially improves all three FMs under shift, without additional computational cost, as shown in Table 13.

Figure 7 shows that as $\beta$ increases, our methods yields larger gains, with DistPFN-T providing additional improvements over DistPFN. For instance, DistPFN and DistPFN-T improve TabICL by 12.3% and 13.7% under $\beta = 5$, respectively. Figure 8 shows the average rank using a critical difference diagram ($\beta = 2$), Furthermore, Table 6 shows consistent gains on three TableShift datasets (Gardner et al., 2023) under label shift. Comparison with ML methods with hyperparameter optimization is provided in Appendix F.

*Table 7.* **Comparison with oracle.** Although our method computes the adjustment factor $\alpha$ using the *predicted* test label distribution, its performance remains close to *DistPFN-Oracle*, which uses the *true* test label ratio.

| $\alpha = \frac{①}{p_{\text{train}}(y)}$ | ① | w/o shift | Shift strength ($\beta$) | | | | | | |
|---|---|---|---|---|---|---|---|---|---|
| | | | 0.0 | 0.1 | 0.5 | 1.0 | 2.0 | 5.0 | Avg. |
| TabPFN-v2 | - | **0.818** | 0.797 | 0.796 | 0.790 | 0.782 | 0.759 | 0.727 | 0.775 |
| + DistPFN | $\widehat{p}_{\text{TabPFN}}(y)$ | **0.818** | 0.799 | 0.797 | 0.795 | 0.791 | 0.783 | 0.769 | 0.789 |
| + DistPFN-T | $\widehat{p}_{\text{DistPFN-T}}(y)$ | **0.818** | 0.799 | 0.798 | 0.797 | 0.796 | 0.789 | 0.775 | 0.792 |
| + DistPFN-Oracle | $p_{\text{test}}(y)$ | **0.818** | **0.803** | **0.802** | **0.800** | **0.797** | **0.792** | **0.784** | **0.796** |

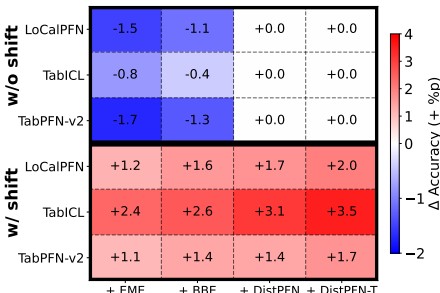

*Figure 9.* Comparison with oracle.

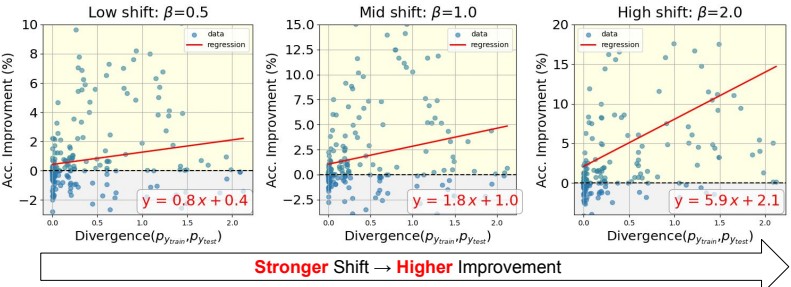

**Stronger** Shift → **Higher** Improvement

*Figure 10.* **Per-dataset improvement.** The figure shows the accuracy improvement for each dataset under varying $\beta$ values with DistPFN-T applied, shown against the KL-divergence between the train and test label distributions of the original dataset.

*Table 8.* Comparison with label shift methods (e.g., EME, BBE).

| Methods | LoCalPFN | | TabICL | | TabPFN-v2 | |
|---|---|---|---|---|---|---|
| | w/o shift | w/ shift | w/o shift | w/ shift | w/o shift | w/ shift |
| - | **0.816** | 0.771 | **0.806** | 0.742 | **0.818** | 0.775 |
| + EME | 0.801 | 0.783 | 0.798 | 0.766 | 0.801 | 0.786 |
| + BBE | 0.805 | 0.787 | 0.802 | 0.768 | 0.805 | 0.789 |
| + DistPFN | **0.816** | 0.788 | **0.806** | 0.773 | **0.818** | 0.789 |
| + DistPFN-T | **0.816** | **0.791** | **0.806** | **0.777** | **0.818** | **0.792** |

*Figure 11.* Comparison with label shift methods (e.g., EME, BBE).

*Table 9.* Predicted distribution: single vs. multiple.

| | Pred. distn. | w/o shift | w/ shift |
|---|---|---|---|
| TabPFN-v2 | - | 0.818 | 0.775 |
| + DistPFN | Single | **0.818** | **0.789** |
| | Multiple | **0.818** | **0.789** |
| + DistPFN-T | Single | **0.818** | 0.791 |
| | Multiple | **0.818** | **0.792** |

# 6. Analysis

In this section, we conduct various analyses on the effectiveness of our methods, DistPFN and DistPFN-T, which are applied to TabPFN-v2 unless otherwise stated.

**Comparison with oracle.** The adjustment factor $\alpha$ in our method is computed using the *predicted* label distribution. We compare this to a variant that uses the *true* label ratio of the test set, referred to as *DistPFN-Oracle*, which is unavailable in practice. This replaces the *predicted* distribution with the *true* distribution as the numerator of $\alpha$. Table 7 and Figure 9 show the results, indicating that our methods achieve performance close to the oracle.

**Performance gain by dataset.** Figure 10 illustrates the accuracy improvement across 253 datasets under three different $\beta$s when DistPFN-T is applied. Each point represents a dataset, with the x-axis showing the KL-divergence between the train and test label distributions of the dataset, and the y-axis indicating the corresponding accuracy improvement. As $\beta$ increases, datasets with larger discrepancies become more imbalanced, with those exhibiting stronger divergence

(i.e., larger shifts) benefiting more from our method.

**Comparison label shift correction methods.** To demonstrate the effectiveness of our methods, we compare it with other techniques handling label shift: EM-based Estimation (**EME**) (Saerens et al., 2002) and Black-box Estimation (**BBE**) (Lipton et al., 2018). Table 8 demonstrates that our method outperforms these approaches without requiring estimation of the test prior. In particular, as shown in Figure 11, while other methods suffer from performance degradation when no shift is present, our method maintains stable performance across both settings.

**Predicted distribution of single vs. multiple instance(s).** As TabPFN allows test instances to be evaluated either individually or in batches, with both yielding identical predictions, DistPFN and DistPFN-T can apply their adjustment based on either the 1) prediction of a *single* instance or 2) the average prediction across *multiple* instances in the test dataset. As shown in Table 9, both choices consistently improve TabPFN-v2, averaged across six $\beta$s for *w/ shift*. The results demonstrate that our method is robust to the choice of distribution source, with full results shown in Appendix L.

*Table 10.* LoCalPFN + Ours.

| $k$ | Methods | Avg. |
|---|---|---|
| 3 | LoCalPFN | 0.750 |
| | + DistPFN | 0.782 |
| | + DistPFN-T | **0.785** |
| 10 | LoCalPFN | 0.770 |
| | + DistPFN | 0.787 |
| | + DistPFN-T | **0.789** |
| 20 | LoCalPFN | 0.771 |
| | + DistPFN | 0.788 |
| | + DistPFN-T | **0.791** |

*Table 11.* **Training prior vs. Training prediction.** Replacing the training prior $p_{\text{train}}(y)$ with the predicted distribution $\widehat{p}_{\text{train}}(y)$ in $\alpha$ shows negligible performance difference, validating the use of $p_{\text{train}}(y)$ as a simple and reliable choice, as $\widehat{p}_{\text{train}}(y)$ requires additional computation.

| $\alpha = \frac{\widehat{p}_{\text{DistPFN(-T)}}(y)}{②}$ | ② | w/o shift | Shift strength ($\beta$) | | | | | | Avg. |
|---|---|---|---|---|---|---|---|---|---|
| | | | 0.0 | 0.1 | 0.5 | 1.0 | 2.0 | 5.0 | |
| TabPFN-v2 | - | 0.818 | 0.797 | 0.796 | 0.790 | 0.782 | 0.759 | 0.727 | 0.775 |
| + DistPFN | $p_{\text{train}}(y)$ | **0.818** | **0.799** | **0.797** | **0.795** | 0.791 | **0.783** | **0.769** | **0.789** |
| | $\widehat{p}_{\text{train}}(y)$ | **0.818** | **0.799** | **0.797** | **0.795** | 0.792 | **0.783** | 0.768 | **0.789** |
| + DistPFN-T | $p_{\text{train}}(y)$ | **0.818** | 0.799 | 0.798 | 0.797 | 0.796 | 0.789 | 0.775 | 0.792 |
| | $\widehat{p}_{\text{train}}(y)$ | **0.818** | **0.800** | **0.800** | **0.798** | **0.797** | **0.791** | **0.777** | **0.793** |

*Table 12.* **Dataset selection with K-means clustering.** Our method remains effective when training subsets are formed by sampling a proportion ($P$) from each of the $K = 10$ clusters, demonstrating robustness to the choice of training subsets.

| $P$ | Methods | Shift strength ($\beta$) | | | | | | Avg. |
|---|---|---|---|---|---|---|---|---|
| | | 0.0 | 0.1 | 0.5 | 1.0 | 2.0 | 5.0 | |
| 0.05 | TabPFN-v2 | 0.644 | 0.639 | 0.591 | 0.547 | 0.505 | 0.460 | 0.554 |
| | + DistPFN | 0.659 | 0.662 | 0.627 | 0.586 | 0.548 | 0.493 | 0.589 |
| | + DistPFN-T | **0.662** | **0.667** | **0.640** | **0.601** | **0.562** | **0.504** | **0.605** |
| 0.10 | TabPFN-v2 | 0.663 | 0.664 | 0.617 | 0.582 | 0.534 | 0.481 | 0.620 |
| | + DistPFN | 0.679 | 0.685 | 0.650 | 0.618 | 0.577 | 0.515 | 0.642 |
| | + DistPFN-T | **0.685** | **0.691** | **0.656** | **0.629** | **0.588** | **0.527** | **0.652** |
| 0.20 | TabPFN-v2 | 0.697 | 0.689 | 0.651 | 0.619 | 0.561 | 0.510 | 0.638 |
| | + DistPFN | 0.713 | 0.711 | 0.682 | 0.663 | 0.610 | 0.553 | 0.676 |
| | + DistPFN-T | **0.716** | **0.715** | **0.689** | **0.670** | **0.620** | **0.563** | **0.688** |

*Table 13.* **Efficiency analysis.** Average prediction time (in seconds) average across 253 datasets with three different backbones.

| | Pred. time | Avg. Acc. |
|---|---|---|
| LoCalPFN | 0.618 | 0.771 |
| + DistPFN | 0.619 | 0.788 |
| + DistPFN-T | 0.619 | **0.791** |
| TabICL | 0.620 | 0.742 |
| + DistPFN | 0.622 | 0.773 |
| + DistPFN-T | 0.622 | **0.777** |
| TabPFN-v2 | 1.002 | 0.775 |
| + DistPFN | 1.003 | 0.789 |
| + DistPFN-T | 1.003 | **0.792** |

**Robustness to $k$ for LoCalPFN.** We apply our methods to LoCalPFN, which improves the efficiency of TabPFN by retrieving $k$ nearest neighbors for each test sample to construct the training dataset. For a fair comparison, we do not fine-tune the model and instead use the pretrained weights of TabPFN-v2, providing a stronger baseline than the original TabPFN used in LoCalPFN. Table 10 reports results across different values of $k$s, averaged over six $\beta$s. The results indicate that our methods consistently improve performance, with full results provided in Appendix J.

**Training prior vs. Training prediction.** The adjustment factor $\alpha$ of our method uses the ground-truth label distribution of the training dataset (i.e., *training prior* or $p_{\text{train}}(y)$) as the denominator, while the numerator is based on the predicted distribution of the test set, introducing a mismatch between true and predicted quantities. To assess the impact of this discrepancy, we analyze an alternative that replaces the training prior with the model's average predicted distribution on the training dataset (i.e., *training prediction* or $\widehat{p}_{\text{train}}(y)$), which requires additional inference. As shown in Table 11, the performance difference is negligible, validating the choice of the training prior.

**Training set selection.** TabPFN suffers from quadratic complexity, making it inefficient for large training datasets (Thomas et al., 2024; Zeng et al., 2025; Qu et al., 2025), and several works mitigate this by training on *selected* subsets. One common approach is to use only the local neighbors of each test sample, as validated in Table 5 with LoCalPFN (Thomas et al., 2024). Another approach clusters the training dataset and selects the centroid and a few nearby samples per cluster. As shown in Table 12, our method remains effective across different percentages of samples per cluster ($P$) under $K = 10$, where $K$ is the number of clusters. Results for various $K$s are provided in Appendix I.

**Efficiency analysis.** To evaluate the efficiency of our method, we compare the average prediction time (seconds) across 253 datasets using TabPFN-v2. Note that our method does not require any additional parameters, and only applies a simple multiplication of an adjustment factor to the predicted results, making it efficient and readily applicable to any tabular foundation model. Table 13 summarizes the results, including the average performance across six $\beta$s, highlighting that our method achieves superior performance gains with negligible computational burden.

*Table 14.* ECE and Precision.

| Method | $\beta$ | Acc. | Prec. | ECE |
|---|---|---|---|---|
| TabPFN-v2 | 2.0 | 0.759 | 0.696 | 0.098 |
| + DistPFN | 2.0 | 0.783 | 0.694 | 0.096 |
| + DistPFN-T | 2.0 | **0.789** | **0.702** | **0.089** |
| TabPFN-v2 | 5.0 | 0.727 | 0.674 | 0.127 |
| + DistPFN | 5.0 | 0.769 | 0.683 | 0.099 |
| + DistPFN-T | 5.0 | **0.775** | **0.695** | **0.093** |

*Table 15.* **Asymmetric cross-entropy (CE).** DistPFN-T employs CE to compute the temperature $\tau$ for temperature scaling, where both directions outperform TabPFN-v2.

| ①: $p_{\text{train}}$, ②: $\widehat{p}_{\text{TabPFN}}(y)$ | | w/o shift | Shift strength ($\beta$) | | | | | | |
|---|---|---|---|---|---|---|---|---|---|
| | | | 0.0 | 0.1 | 0.5 | 1.0 | 2.0 | 5.0 | Avg. |
| TabPFN-v2 | | **0.818** | 0.797 | 0.796 | 0.790 | 0.782 | 0.759 | 0.727 | 0.775 |
| + DistPFN-T | $\tau = \text{CE}(①, ②)$ | **0.818** | **0.799** | **0.799** | **0.797** | 0.795 | 0.788 | 0.769 | 0.791 |
| | $\tau = \text{CE}(②, ①)$ | **0.818** | **0.799** | 0.798 | **0.797** | **0.796** | **0.789** | **0.775** | **0.792** |

*Table 16.* DistPFN-T CE-based vs. CV-based $\tau$.

| $\beta$ | 0.0 | 0.5 | 1.0 | 2.0 | 5.0 |
|---|---|---|---|---|---|
| TabPFN-v2 | 0.797 | 0.790 | 0.782 | 0.759 | 0.727 |
| + DistPFN-T (CV) | 0.779 | 0.792 | 0.791 | 0.785 | 0.761 |
| + DistPFN-T (CE) | **0.799** | **0.797** | **0.796** | **0.789** | **0.775** |

*Table 17.* $\tau$ statistics.

| Metric | Median | Mean |
|---|---|---|
| CE | 1.004 | 1.339 |
| KL | 0.162 | 0.497 |
| JS | 0.040 | 0.090 |
| L2 | 0.334 | 0.396 |

*Table 18.* $\tau$ metric ablation.

| $\beta$ | 0.0 | 0.5 | 1.0 | 2.0 | 5.0 |
|---|---|---|---|---|---|
| KL | .649 | .683 | .690 | .694 | .696 |
| JS | .614 | .622 | .629 | .636 | .641 |
| L2 | .788 | .785 | .778 | .774 | .761 |
| CE | **.799** | **.797** | **.796** | **.789** | **.775** |

*Table 19.* DistPFN applied to tree-based models.

| $\beta$ | 0.0 | 0.1 | 0.5 | 1.0 | 2.0 | 5.0 |
|---|---|---|---|---|---|---|
| XGBoost | 0.763 | 0.759 | 0.743 | 0.715 | 0.664 | 0.623 |
| + DistPFN | **0.770** | **0.768** | **0.758** | 0.742 | 0.703 | 0.672 |
| + DistPFN-T | 0.768 | **0.768** | **0.758** | **0.743** | **0.710** | **0.679** |
| RandomForest | 0.769 | 0.767 | 0.750 | 0.720 | 0.664 | 0.617 |
| + DistPFN | **0.776** | **0.776** | 0.771 | 0.764 | 0.742 | 0.717 |
| + DistPFN-T | **0.776** | **0.776** | **0.772** | **0.768** | **0.752** | **0.730** |
| LightGBM | 0.758 | 0.753 | 0.734 | 0.705 | 0.657 | 0.618 |
| + DistPFN | **0.764** | **0.761** | **0.748** | 0.727 | 0.682 | 0.645 |
| + DistPFN-T | **0.764** | **0.761** | **0.748** | **0.728** | **0.687** | **0.650** |

**Calibration and precision analysis.** Beyond accuracy, we evaluate whether DistPFN improves model calibration. Table 14 reports expected calibration error (ECE) and precision across 253 datasets, averaged over 5 seeds. DistPFN-T reduces ECE and improves precision, confirming that it benefits both calibration and discriminative performance.

**Asymmetric cross-entropy.** DistPFN-T employs cross-entropy (CE) to compute the $\tau$ for temperature scaling, where the value differs depending on whether it is computed as $\text{CE}(\widehat{p}_{\text{TabPFN}}(y), p_{\text{train}})$ or $\text{CE}(p_{\text{train}}, \widehat{p}_{\text{TabPFN}}(y))$. Table 15 shows that both directions outperform TabPFN-v2, demonstrating robustness to the choice of direction.

**Comparison with CV-based temperature tuning.** We compare DistPFN-T against a cross-validation baseline that selects the optimal temperature $\tau$ from {0.01, 0.05, 0.1, 0.2, 0.5, 1.0, 2.0, 5.0, 10.0} using 3-fold CV, averaged over 5 seeds across 253 datasets. As shown in Table 16, DistPFN-T (CE) outperforms the CV-based approach while being more efficient, requiring only a single forward pass.

$\tau$ **metric ablation.** We compare four metrics for computing the temperature $\tau$ in DistPFN-T: cross-entropy, KL divergence, Jensen–Shannon divergence, and L2 distance. Table 17 shows the statistics of $\tau$ across 253 datasets at

$\beta = 2.0$, and Table 18 reports the accuracy, where CE achieves the best performance. The gap appears to be largely driven by scale suitability, where KL and JS produce near-zero $\tau$ values, while CE naturally produces $\tau \approx 1.0$.

**Application to tree-based models.** Although DistPFN is designed for ICL-based models (see Table 2), it can be applied to any model that outputs class probabilities. As shown in Table 19, we evaluate DistPFN on XGBoost (Chen & Guestrin, 2016), RandomForest (Liaw & Wiener, 2002), and LightGBM (Ke et al., 2017), averaged over 253 datasets and 5 seeds. DistPFN-T yields consistent improvements even for tree-based models, with gains increasing as $\beta$ grows.

# 7. Conclusion

In this work, we introduce DistPFN, a test-time adjustment method to mitigate label shift in tabular FMs using ICL. We further propose DistPFN-T to stabilize the adjustment via temperature scaling based on distributional divergence between training prior and predicted distribution.

**Limitations and future works.** While our method effectively handles label shift without retraining, it does not address feature shift, which can also occur in practice. A potential direction for future work is to design FMs that are inherently robust to both label and feature shift, beyond post-hoc adjustment. In addition, beyond the TabPFN-based property of directly referencing the training dataset at inference time, future work may explore shift-handling strategies that more explicitly account for model-specific characteristics. We hope that this work encourages further exploration of robustness to distribution shifts in tabular ICL.

## Impact Statement

This work proposes a post-hoc adjustment method for tabular foundation models under label shift. Our method does not introduce new data collection, model training, or deployment mechanisms; it operates on existing pretrained models at inference time. We do not foresee specific negative societal impacts beyond those generally associated with machine learning for tabular data. By improving robustness under class distribution shift, our work may contribute to fairer predictions in applications where class imbalance is prevalent, such as healthcare and finance.

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

# Appendix

## A. Experimental Setups

**Experimental setups.** We use the official implementation of TabPFN[2] and adopt all default settings without modification. This includes architectural choices such as the number of layers and hidden dimensions, where we use 12 transformer layers, each with a hidden size of 192 and 6 attention heads. The feedforward layer dimension is implicitly set to 768 via a hidden factor of 4. For inference, we load the pretrained weights from TabPFN-v2[3] available on Hugging Face.

**Dataset.** We evaluate on 250+ tabular datasets from OpenML (Bischl et al., 2017). The dataset list is retrieved from the benchmark configuration provided in this repository[4], which is built on top of the official TabPFN evaluation setup. Dataset statistics are summarized in Appendix H.

## B. Baseline Methods

We categorize 15 baseline tabular models into three groups:

- **ML models (7):** Logistic Regression (LR), Support Vector Machines (SVM), Random Forest (Liaw & Wiener, 2002), $k$-nearest neighbors ($k$NN), Multi-layer Perceptrons (MLP), LightGBM (Ke et al., 2017), CatBoost (Prokhorenkova et al., 2018)
- **DL (non-foundation) models (5):** FT-Transformer (Gorishniy et al., 2021), TabM (Gorishniy et al., 2024a), TabulaRNN (Thielmann & Samiee, 2024), MambaTab (Ahamed & Cheng, 2024), RealMLP (Holzmüller et al., 2024)
- **DL (foundation) models based on ICL (3):** TabPFN-v2 (Hollmann et al., 2025), LoCalPFN (Thomas et al., 2024), TabICL (Qu et al., 2025)

Details of each method are provided below.

### B.1. Machine Learning (ML) Models

- **Logistic Regression (LR)** (Cox, 1958): A simple linear model commonly used for binary and multiclass classification tasks in tabular data.
- **Support Vector Machine (SVM)** (Cortes & Vapnik, 1995): A kernel-based classifier that aims to find the optimal decision boundary with maximum margin between classes.
- **Multilayer Perceptron (MLP)** (Haykin, 1994): A feedforward neural network consisting of multiple fully connected layers with non-linear activations, trained via backpropagation.
- **$k$-Nearest Neighbors** ($k$NN) (Altman, 1992): A non-parametric method that classifies a sample based on the majority class among its $k$ nearest neighbors in the feature space.
- **Random Forest** (Liaw & Wiener, 2002): An ensemble learning method based on bagging over decision trees, which improves robustness and generalization.
- **LightGBM** (Ke et al., 2017): A fast and efficient GBDT model using histogram-based algorithms and leaf-wise tree growth.
- **CatBoost** (Prokhorenkova et al., 2018): A GBDT model that handles categorical features efficiently and mitigates prediction shift via ordered boosting.

---

[2]https://github.com/PriorLabs/TabPFN
[3]https://huggingface.co/Prior-Labs/TabPFN-v2-clf
[4]https://github.com/carteakey/tabpfn-eval/blob/main/src/data/openml_list.csv

**B.2. Deep Learning (Non-Foundation) Models**

- **FT-Transformer (Gorishniy et al., 2021)**: A transformer-based architecture tailored for tabular data, providing a simple yet powerful baseline that outperforms many prior DL models on classification and regression tasks.

- **TabM (Gorishniy et al., 2024a)**: An MLP-based model that leverages an efficient ensemble mechanism to approximate deep ensembles, enabling multiple predictions per instance while maintaining computational efficiency.

- **TabulaRNN (Thielmann & Samiee, 2024)**: An RNN-inspired architecture for tabular data that emphasizes efficiency, addressing limitations of NLP-style models in terms of scalability and training cost.

- **MambaTab (Ahamed & Cheng, 2024)**: A scalable and efficient model built on structured state-space models (SSMs), capturing long-range dependencies with fewer parameters while maintaining strong predictive performance.

- **RealMLP (Holzmüller et al., 2024)**: An enhanced MLP variant with meta-tuned hyperparameters, achieving competitive accuracy–efficiency trade-offs compared to gradient boosting methods in tabular benchmarks.

**B.3. Deep Learning (Foundation) Models based on ICL**

- **LoCalPFN (Thomas et al., 2024)**: A lightweight PFN variant that reduces computational cost by leveraging local task priors and architectural simplifications.

- **TabICL (Qu et al., 2025)**: A two-stage model that first applies column attention to capture feature dependencies and then row attention to encode sample interactions.

- **TabPFN-v2 (Hollmann et al., 2025)**: A state-of-the-art foundation model for tabular classification that leverages a pretrained transformer for zero-shot prediction on small datasets.

## C. Baseline Implementations

The baseline results are obtained from the following publicly available repositories:

- [1] **TabPFN Evaluation** framework[5] was used to evaluate all ML models, as well as the foundation models *TabPFN* and *LoCalPFN*. Since *LoCalPFN* does not have an official implementation, we reimplemented it based on the TabPFN codebase.

- [2] **AutoGluon v1.4.0**[6] was used to benchmark *TabICL* and several non-foundation models such as *FT-Transformer*, *TabM*, and *RealMLP*.

- [3] **Mambular**[7] provided implementations for additional non-foundation models including *MambaTab*, *TabulaRNN*, *FT-Transformer*, and *TabM*.

For models implemented in both [2] and [3] (e.g., FT-Transformer and TabM), we use the [2] versions as they yield stronger performance for a stronger baseline.

---

[5] https://github.com/carteakey/tabpfn-eval
[6] https://auto.gluon.ai/
[7] https://github.com/OpenTabular/DeepTabular

# D. Theoretical Justification

We provide theoretical grounding for our posterior adjustment to clarify that DistPFN is not merely a heuristic trick, but a principled approximation derived from existing theory. We present two complementary perspectives: 1) connection to classical label shift correction as a plug-in reweighting (Section D.1) and 2) Bayesian view that replaces the mismatched prior with a self-consistent estimate from model predictions (Section D.2).

## D.1. Relation to Label Shift Correction

The label shift setting assumes that the conditional distribution $p(x|y)$ remains invariant while the marginal priors differ:

$$p_{\text{train}}(y) \neq p_{\text{test}}(y), \quad p(x|y) \text{ is fixed.}$$

Under this assumption, the Bayes-optimal posterior is given by

$$p_{\text{test}}(y|x) \propto \frac{p_{\text{train}}(y|x)}{p_{\text{train}}(y)} \, p_{\text{test}}(y).$$

Classical approaches such as EM-based reweighting (Saerens et al., 2002; Lipton et al., 2018) estimate $p_{\text{test}}(y)$ explicitly by matching marginal predictions to unlabeled test data. DistPFN instead uses the predictive marginal $\hat{p}(y)$ obtained directly from the model, and constructs the adjustment factor

$$\alpha(y) = \frac{\hat{p}(y)}{p_{\text{train}}(y)}.$$

This yields the corrected posterior

$$\hat{p}(y|x) \propto \frac{p_{\text{train}}(y|x)}{p_{\text{train}}(y)} \, \hat{p}(y),$$

which can be seen as a plug-in realization of the classical correction rule, avoiding iterative estimation while remaining theoretically consistent with label shift correction.

## D.2. Bayesian Interpretation

From a Bayesian perspective, TabPFN models the posterior under the training distribution:

$$p_{\text{train}}(y|x) \propto p(x|y) \, p_{\text{train}}(y).$$

At test time, however, the desired posterior is

$$p_{\text{test}}(y|x) \propto p(x|y) \, p_{\text{test}}(y).$$

The difference comes solely from the prior. DistPFN addresses this gap by substituting $p_{\text{train}}(y)$ with $\hat{p}(y)$, the average predictive distribution obtained on the test set:

$$\hat{p}_{\text{DistPFN}}(y|x) \propto \frac{p_{\text{train}}(y|x)}{p_{\text{train}}(y)} \, \hat{p}(y).$$

This interpretation shows that DistPFN is not an ad-hoc adjustment but a Bayesian posterior correction where the unknown test prior is approximated in a self-consistent manner from model outputs. The method therefore inherits a principled justification while retaining the efficiency of a simple, training-free plug-in procedure.

# E. Classical Methods for Label Shift Correction

In this section, we summarize three representative approaches for handling label shift. All of these methods directly adjust classifier outputs, but they differ in how the test prior $\pi_{\text{test}}$ is obtained.

## E.1. Prior-ratio Adjustment

Prior-ratio Adjustment (Elkan, 2001) introduces a simple correction under changing class priors in the binary setting. The method assumes that the new prior $\pi_{\text{test}}$ is available from external knowledge or domain statistics. Given a posterior $p_{\text{train}}(1|x)$ trained under $\pi_{\text{train}}$, the corrected posterior is

$$p_{\text{test}}(1|x) \;=\; \frac{p_{\text{train}}(1|x) \cdot \frac{\pi_{\text{test}}(1)}{\pi_{\text{train}}(1)}}{p_{\text{train}}(1|x) \cdot \frac{\pi_{\text{test}}(1)}{\pi_{\text{train}}(1)} + (1 - p_{\text{train}}(1|x)) \cdot \frac{1 - \pi_{\text{test}}(1)}{1 - \pi_{\text{train}}(1)}}.$$

This approach directly modifies posterior probabilities by scaling them with prior ratios. The same principle naturally extends to the multiclass case by applying the ratio $\pi_{\text{test}}(y)/\pi_{\text{train}}(y)$ to each class posterior.

## E.2. EM-based Estimation

EM-based Estimation (Saerens et al., 2002) proposes an iterative procedure to estimate unknown test priors when they are not directly given. At iteration $t$, the posterior is updated by

$$p^{(t+1)}(y|x) \;\propto\; p_{\text{train}}(y|x) \cdot \frac{\pi_{\text{test}}^{(t)}(y)}{\pi_{\text{train}}(y)}.$$

The updated posteriors provide a new estimate of $\pi_{\text{test}}$ by averaging across the test set. Repeating this E-step and M-step allows the estimated test prior to gradually converge. The final corrected posterior then follows the standard prior-ratio adjustment, but with $\pi_{\text{test}}$ estimated rather than assumed.

## E.3. Black-box Estimation

Black-box Estimation (Lipton et al., 2018) employs a validation dataset with true labels to construct a confusion matrix $C(s|y) = P(\hat{y} = s \mid y)$ that characterizes prediction errors of the classifier. On an unlabeled test set, it collects predicted labels to obtain the empirical distribution $p_{\text{test}}(s)$. These quantities are related through the equation

$$p_{\text{test}}(s) \;\approx\; \sum_y C(s|y)\, \pi_{\text{test}}(y).$$

By solving this linear system, the method estimates the test prior $\pi_{\text{test}}$. Once the test prior is recovered, the posterior correction is applied using the prior ratio:

$$p_{\text{test}}(y|x) \;\propto\; p_{\text{train}}(y|x) \cdot \frac{\pi_{\text{test}}(y)}{\pi_{\text{train}}(y)}.$$

This approach is considered black-box as it does not require access to classifier internals, only its predicted outputs and a validation set to estimate the confusion matrix.

# F. Full Results

*Table F.1.* **Tabular classification results.** While most baselines suffer substantial performance degradation under label shift, our methods significantly improve the accuracy of tabular FMs (e.g., TabPFN-v2) across varying degrees of shift ($\beta$), averaged over 253 datasets.

| | Methods | w/o shift | Shift strength ($\beta$) 0.0 | 0.1 | 0.5 | 1.0 | 2.0 | 5.0 | Avg. |
|---|---|---|---|---|---|---|---|---|---|
| **Machine Learning** | LogReg. | $0.765_{\pm 0.002}$ | $0.719_{\pm 0.002}$ | $0.709_{\pm 0.002}$ | $0.674_{\pm 0.004}$ | $0.634_{\pm 0.004}$ | $0.597_{\pm 0.002}$ | $0.566_{\pm 0.004}$ | 0.650 |
| | + HPO | $0.771_{\pm 0.003}$ | $0.697_{\pm 0.005}$ | $0.687_{\pm 0.003}$ | $0.653_{\pm 0.003}$ | $0.616_{\pm 0.006}$ | $0.586_{\pm 0.003}$ | $0.550_{\pm 0.003}$ | 0.631 |
| | SVM | $0.780_{\pm 0.003}$ | $0.684_{\pm 0.002}$ | $0.646_{\pm 0.004}$ | $0.560_{\pm 0.005}$ | $0.531_{\pm 0.003}$ | $0.486_{\pm 0.004}$ | $0.448_{\pm 0.004}$ | 0.559 |
| | + HPO | $0.784_{\pm 0.003}$ | $0.731_{\pm 0.001}$ | $0.689_{\pm 0.008}$ | $0.626_{\pm 0.003}$ | $0.597_{\pm 0.005}$ | $0.570_{\pm 0.005}$ | $0.541_{\pm 0.007}$ | 0.626 |
| | MLP | $0.778_{\pm 0.004}$ | $0.658_{\pm 0.005}$ | $0.647_{\pm 0.004}$ | $0.613_{\pm 0.005}$ | $0.574_{\pm 0.006}$ | $0.527_{\pm 0.004}$ | $0.493_{\pm 0.001}$ | 0.585 |
| | + HPO | $0.795_{\pm 0.005}$ | $0.706_{\pm 0.006}$ | $0.688_{\pm 0.006}$ | $0.654_{\pm 0.008}$ | $0.615_{\pm 0.006}$ | $0.580_{\pm 0.005}$ | $0.545_{\pm 0.005}$ | 0.631 |
| | $k$NN | $0.765_{\pm 0.004}$ | $0.663_{\pm 0.004}$ | $0.657_{\pm 0.004}$ | $0.629_{\pm 0.003}$ | $0.589_{\pm 0.003}$ | $0.538_{\pm 0.003}$ | $0.501_{\pm 0.004}$ | 0.596 |
| | + HPO | $0.783_{\pm 0.002}$ | $0.693_{\pm 0.002}$ | $0.684_{\pm 0.003}$ | $0.644_{\pm 0.004}$ | $0.588_{\pm 0.003}$ | $0.540_{\pm 0.003}$ | $0.498_{\pm 0.002}$ | 0.608 |
| | Random Forest | $0.796_{\pm 0.003}$ | $0.768_{\pm 0.003}$ | $0.765_{\pm 0.003}$ | $0.748_{\pm 0.005}$ | $0.718_{\pm 0.004}$ | $0.665_{\pm 0.005}$ | $0.618_{\pm 0.005}$ | 0.714 |
| | + HPO | $0.803_{\pm 0.002}$ | $0.771_{\pm 0.002}$ | $0.767_{\pm 0.001}$ | $0.743_{\pm 0.004}$ | $0.701_{\pm 0.004}$ | $0.627_{\pm 0.008}$ | $0.578_{\pm 0.006}$ | 0.698 |
| | LightGBM | $0.789_{\pm 0.003}$ | $0.758_{\pm 0.004}$ | $0.753_{\pm 0.002}$ | $0.734_{\pm 0.003}$ | $0.705_{\pm 0.004}$ | $0.657_{\pm 0.005}$ | $0.618_{\pm 0.005}$ | 0.704 |
| | + HPO | $0.790_{\pm 0.006}$ | $0.726_{\pm 0.008}$ | $0.661_{\pm 0.005}$ | $0.655_{\pm 0.008}$ | $0.608_{\pm 0.008}$ | $0.577_{\pm 0.015}$ | $0.551_{\pm 0.004}$ | 0.630 |
| | CatBoost | $0.803_{\pm 0.001}$ | $0.774_{\pm 0.002}$ | $0.771_{\pm 0.002}$ | $0.751_{\pm 0.004}$ | $0.718_{\pm 0.004}$ | $0.665_{\pm 0.005}$ | $0.621_{\pm 0.005}$ | 0.717 |
| | + HPO | $0.802_{\pm 0.002}$ | $0.774_{\pm 0.002}$ | $0.771_{\pm 0.002}$ | $0.752_{\pm 0.004}$ | $0.719_{\pm 0.004}$ | $0.665_{\pm 0.006}$ | $0.621_{\pm 0.005}$ | 0.717 |
| **Deep Learning** — Non-FMs | FT-Transformer | $0.784_{\pm 0.002}$ | $0.748_{\pm 0.004}$ | $0.746_{\pm 0.004}$ | $0.718_{\pm 0.005}$ | $0.674_{\pm 0.005}$ | $0.610_{\pm 0.003}$ | $0.551_{\pm 0.007}$ | 0.675 |
| | TabM | $0.794_{\pm 0.002}$ | $0.762_{\pm 0.004}$ | $0.757_{\pm 0.004}$ | $0.735_{\pm 0.003}$ | $0.694_{\pm 0.005}$ | $0.624_{\pm 0.006}$ | $0.565_{\pm 0.006}$ | 0.690 |
| | TabulaRNN | $0.749_{\pm 0.003}$ | $0.699_{\pm 0.003}$ | $0.684_{\pm 0.003}$ | $0.641_{\pm 0.004}$ | $0.585_{\pm 0.009}$ | $0.522_{\pm 0.011}$ | $0.465_{\pm 0.008}$ | 0.599 |
| | MambaTab | $0.719_{\pm 0.004}$ | $0.629_{\pm 0.006}$ | $0.603_{\pm 0.004}$ | $0.525_{\pm 0.002}$ | $0.466_{\pm 0.010}$ | $0.430_{\pm 0.005}$ | $0.394_{\pm 0.002}$ | 0.508 |
| | RealMLP | $0.794_{\pm 0.002}$ | $0.760_{\pm 0.004}$ | $0.758_{\pm 0.005}$ | $0.745_{\pm 0.003}$ | $0.720_{\pm 0.005}$ | $0.677_{\pm 0.002}$ | $0.643_{\pm 0.004}$ | 0.717 |
| **FMs** | LoCalPFN | $\mathbf{0.816}_{\pm 0.002}$ | $0.794_{\pm 0.003}$ | $0.793_{\pm 0.004}$ | $0.788_{\pm 0.003}$ | $0.778_{\pm 0.002}$ | $0.753_{\pm 0.004}$ | $0.719_{\pm 0.000}$ | 0.771 |
| | + DistPFN | $\mathbf{0.816}_{\pm 0.002}$ | $\underline{0.797}_{\pm 0.001}$ | $\underline{0.796}_{\pm 0.002}$ | $\underline{0.794}_{\pm 0.002}$ | $\underline{0.790}_{\pm 0.002}$ | $\underline{0.782}_{\pm 0.001}$ | $\underline{0.770}_{\pm 0.003}$ | $\underline{0.788}$ |
| | + DistPFN-T | $\mathbf{0.816}_{\pm 0.002}$ | $\mathbf{0.798}_{\pm 0.002}$ | $\mathbf{0.797}_{\pm 0.002}$ | $\mathbf{0.796}_{\pm 0.002}$ | $\mathbf{0.794}_{\pm 0.002}$ | $\mathbf{0.787}_{\pm 0.001}$ | $\mathbf{0.776}_{\pm 0.003}$ | $\mathbf{0.791}$ |
| | TabICL | $\mathbf{0.806}_{\pm 0.002}$ | $0.783_{\pm 0.003}$ | $0.781_{\pm 0.003}$ | $0.770_{\pm 0.003}$ | $0.747_{\pm 0.003}$ | $0.704_{\pm 0.006}$ | $0.664_{\pm 0.006}$ | 0.742 |
| | + DistPFN | $\mathbf{0.806}_{\pm 0.002}$ | $\underline{0.786}_{\pm 0.002}$ | $\underline{0.786}_{\pm 0.002}$ | $\underline{0.781}_{\pm 0.002}$ | $\underline{0.776}_{\pm 0.002}$ | $\underline{0.763}_{\pm 0.002}$ | $\underline{0.746}_{\pm 0.004}$ | $\underline{0.773}$ |
| | + DistPFN-T | $\mathbf{0.806}_{\pm 0.003}$ | $\mathbf{0.786}_{\pm 0.003}$ | $\mathbf{0.786}_{\pm 0.003}$ | $\mathbf{0.783}_{\pm 0.002}$ | $\mathbf{0.780}_{\pm 0.002}$ | $\mathbf{0.771}_{\pm 0.001}$ | $\mathbf{0.755}_{\pm 0.004}$ | $\mathbf{0.777}$ |
| | TabPFN-v2 | $\mathbf{0.818}_{\pm 0.004}$ | $\underline{0.797}_{\pm 0.003}$ | $0.796_{\pm 0.004}$ | $0.790_{\pm 0.002}$ | $0.782_{\pm 0.002}$ | $0.759_{\pm 0.003}$ | $0.727_{\pm 0.003}$ | 0.775 |
| | + DistPFN | $\mathbf{0.818}_{\pm 0.002}$ | $\mathbf{0.799}_{\pm 0.001}$ | $\underline{0.797}_{\pm 0.002}$ | $\underline{0.795}_{\pm 0.002}$ | $\underline{0.791}_{\pm 0.003}$ | $\underline{0.783}_{\pm 0.003}$ | $\underline{0.769}_{\pm 0.003}$ | $\underline{0.789}$ |
| | + DistPFN-T | $\mathbf{0.818}_{\pm 0.002}$ | $\mathbf{0.799}_{\pm 0.003}$ | $\mathbf{0.798}_{\pm 0.002}$ | $\mathbf{0.797}_{\pm 0.002}$ | $\mathbf{0.796}_{\pm 0.003}$ | $\mathbf{0.789}_{\pm 0.003}$ | $\mathbf{0.775}_{\pm 0.003}$ | $\mathbf{0.792}$ |

# G. Hyperparameter Tuning for Stronger Baselines

To ensure strong and fair baselines, we perform hyperparameter tuning for each conventional ML method using the search space provided in a public implementation[8]. The search spaces are manually designed to cover commonly used ranges for each model class, including both optimization-related parameters and regularization or structural options. We conduct random search over these spaces and tune the models on validation datasets that are kept separate from the final test splits. The details of the hyperparameter search spaces are provided in Table G.1.

*Table G.1.* Hyperparameter search spaces for each conventional ML baseline.

| Model | Hyperparameter | Type | Log-scale | Range |
|---|---|---|---|---|
| Logistic Regression | max_iter | int | no | {50, 100, 200, 500, 1000} |
| | solver | categorical | no | {newton-cg, lbfgs, liblinear, sag, saga} |
| | fit_intercept | boolean | no | {True, False} |
| | penalty | categorical | no | {l1, l2, elasticnet, none} |
| | C | float | no | {0.1, 1.0, 10.0, 100.0} |
| Random Forest | n_estimators | int | no | {10, 50, 100, 200, 500} |
| | criterion | categorical | no | {gini, entropy} |
| | probability | boolean | no | {True} |
| | max_depth | int / None | no | {None, 10, 50, 100, 200} |
| | min_samples_split | int | no | {2, 5, 10} |
| | min_samples_leaf | int | no | {1, 2, 4} |
| | max_features | categorical | no | {auto, sqrt, log2} |
| SVM | C | float | no | {0.1, 1.0, 10.0, 100.0} |
| | kernel | categorical | no | {linear, poly, rbf, sigmoid} |
| | probability | boolean | no | {True} |
| | degree | int | no | {2, 3, 4, 5} |
| | gamma | categorical | no | {scale, auto} |
| MLP | max_iter | int | no | {50, 100, 200, 500, 1000} |
| | activation | categorical | no | {identity, logistic, tanh, relu} |
| | solver | categorical | no | {lbfgs, sgd, adam} |
| | alpha | float | no | {0.0001, 0.001, 0.01, 0.1} |
| | learning_rate | categorical | no | {constant, invscaling, adaptive} |
| | learning_rate_init | float | no | {0.001, 0.01, 0.1} |
| kNN | n_neighbors | int | no | {3, 5, 11, 19} |
| | weights | categorical | no | {uniform, distance} |
| | algorithm | categorical | no | {auto, ball_tree, kd_tree, brute} |
| | leaf_size | int | no | {30, 50, 100} |
| | p | int | no | {1, 2} |
| XGBoost | n_estimators | int | no | {50, 100, 200} |
| | max_depth | int | no | {6, 10, 15, 20} |
| | learning_rate | float | no | {0.001, 0.01, 0.1} |
| | subsample | float | no | {0.5, 0.6, 0.7, 0.8, 0.9, 1.0} |
| | colsample_bytree | float | no | {0.4, 0.5, ..., 1.0} |
| | colsample_bylevel | float | no | {0.4, 0.5, ..., 1.0} |
| LightGBM | n_estimators | int | no | {50, 100, 200} |
| | max_depth | int | no | {6, 10, 15, 20} |
| | learning_rate | float | no | {0.001, 0.01, 0.1} |
| | num_leaves | int | no | {31, 60, 120, 240, 480, 960} |
| | min_child_samples | int | no | {10, 20, 30, 40, 50} |
| CatBoost | iterations | int | no | {50, 100, 200} |
| | depth | int | no | {6, 8, 10} |
| | learning_rate | float | no | {0.001, 0.01, 0.1} |
| | l2_leaf_reg | float | no | {1, 3, 5, 7, 9} |

---

[8] https://github.com/carteakey/tabpfn-eval

# H. Dataset Statistics

We evaluate on 253 tabular datasets from OpenML (Bischl et al., 2017). Summary statistics for all datasets are provided in Table H.1, H.2, and H.3. Each dataset is described using the following attributes: the dataset name (**Name**), the total number of input features (**#Features**), the number of categorical features among them (**#Cat. Feat.**), the number of data instances (**#Instances**), the number of class labels (**#Classes**), the number of missing values (**#NaNs**), and the number of samples belonging to the smallest class (**Minority Class Size**).

*Table H.1.* Dataset statistics - Part 1

| Name | #Features | #Cat. Feat. | #Instances | #Classes | #NaNs | Minority Class Size |
|---|---|---|---|---|---|---|
| pollen | 6 | 1 | 3848 | 2 | 0 | 1924 |
| Sick_numeric | 30 | 1 | 3772 | 2 | 0 | 231 |
| jungle_chess_2pcs_endgame_rat_rat | 47 | 27 | 3660 | 2 | 0 | 1605 |
| UCI_churn | 21 | 1 | 3333 | 2 | 0 | 483 |
| led24 | 25 | 25 | 3200 | 10 | 0 | 296 |
| led7 | 8 | 8 | 3200 | 10 | 0 | 270 |
| kr-vs-kp | 37 | 37 | 3196 | 2 | 0 | 1527 |
| splice | 61 | 61 | 3190 | 3 | 0 | 767 |
| space_ga | 7 | 1 | 3107 | 2 | 0 | 1541 |
| StackOverflow-polarity-train | 2 | 1 | 3097 | 3 | 0 | 842 |
| seismic-bumps | 19 | 5 | 2584 | 2 | 0 | 170 |
| ozone-level-8hr | 73 | 1 | 2534 | 2 | 0 | 160 |
| jungle_chess_2pcs_endgame_lion_lion | 47 | 27 | 2352 | 2 | 0 | 949 |
| jungle_chess_2pcs_endgame_elephant_elephant | 47 | 27 | 2351 | 2 | 0 | 1035 |
| segment | 20 | 1 | 2310 | 7 | 0 | 330 |
| Titanic | 4 | 1 | 2201 | 2 | 0 | 711 |
| quake | 4 | 1 | 2178 | 2 | 0 | 969 |
| kc1 | 22 | 1 | 2109 | 2 | 0 | 326 |
| balloon | 2 | 1 | 2001 | 2 | 0 | 482 |
| mfeat-fourier | 77 | 1 | 2000 | 10 | 0 | 200 |
| ozone-level-8hr_seed_0_nrows_2000_nclasses_10_ncols_100_stratify_True | 73 | 1 | 2000 | 2 | 0 | 126 |
| mfeat-karhunen | 65 | 1 | 2000 | 10 | 0 | 200 |
| jannis_seed_0_nrows_2000_nclasses_10_ncols_100_stratify_True | 55 | 1 | 2000 | 2 | 0 | 1000 |
| covertype_seed_0_nrows_2000_nclasses_10_ncols_100_stratify_True | 55 | 45 | 2000 | 2 | 0 | 1000 |
| first-order-theorem-proving_seed_0_nrows_2000_nclasses_10_ncols_100_stratify_True | 52 | 1 | 2000 | 6 | 0 | 159 |
| MiniBooNE_seed_0_nrows_2000_nclasses_10_ncols_100_stratify_True | 51 | 1 | 2000 | 2 | 0 | 1000 |
| KDDCup09_upselling_seed_0_nrows_2000_nclasses_10_ncols_100_stratify_True | 50 | 16 | 2000 | 2 | 0 | 1000 |
| ada_seed_0_nrows_2000_nclasses_10_ncols_100_stratify_True | 49 | 1 | 2000 | 2 | 0 | 496 |
| mfeat-zernike | 48 | 1 | 2000 | 10 | 0 | 200 |
| connect-4_seed_0_nrows_2000_nclasses_10_ncols_100_stratify_True | 43 | 43 | 2000 | 3 | 0 | 191 |
| kr-vs-kp_seed_0_nrows_2000_nclasses_10_ncols_100_stratify_True | 37 | 37 | 2000 | 2 | 0 | 956 |
| road-safety_seed_0_nrows_2000_nclasses_10_ncols_100_stratify_True | 33 | 4 | 2000 | 2 | 0 | 1000 |
| GesturePhaseSegmentationProcessed_seed_0_nrows_2000_nclasses_10_ncols_100_stratify_True | 33 | 1 | 2000 | 5 | 0 | 202 |
| PhishingWebsites_seed_0_nrows_2000_nclasses_10_ncols_100_stratify_True | 31 | 31 | 2000 | 2 | 0 | 886 |
| pol_seed_0_nrows_2000_nclasses_10_ncols_100_stratify_True | 27 | 1 | 2000 | 2 | 0 | 1000 |
| Higgs_seed_0_nrows_2000_nclasses_10_ncols_100_stratify_True | 25 | 1 | 2000 | 2 | 0 | 1000 |
| eye_movements_seed_0_nrows_2000_nclasses_10_ncols_100_stratify_True | 24 | 4 | 2000 | 2 | 0 | 1000 |
| numerai28.6_seed_0_nrows_2000_nclasses_10_ncols_100_stratify_True | 22 | 1 | 2000 | 2 | 0 | 990 |
| kc1_seed_0_nrows_2000_nclasses_10_ncols_100_stratify_True | 22 | 1 | 2000 | 2 | 0 | 309 |
| kdd_ipums_la_97-small_seed_0_nrows_2000_nclasses_10_ncols_100_stratify_True | 21 | 1 | 2000 | 2 | 0 | 1000 |
| churn_seed_0_nrows_2000_nclasses_10_ncols_100_stratify_True | 21 | 5 | 2000 | 2 | 0 | 283 |
| compass_seed_0_nrows_2000_nclasses_10_ncols_100_stratify_True | 18 | 10 | 2000 | 2 | 0 | 1000 |
| house_16H_seed_0_nrows_2000_nclasses_10_ncols_100_stratify_True | 17 | 1 | 2000 | 2 | 0 | 1000 |
| segment_seed_0_nrows_2000_nclasses_10_ncols_100_stratify_True | 17 | 1 | 2000 | 7 | 0 | 285 |
| adult_seed_0_nrows_2000_nclasses_10_ncols_100_stratify_True | 15 | 9 | 2000 | 2 | 242 | 479 |
| adult_seed_1_nrows_2000_nclasses_10_ncols_100_stratify_True | 15 | 9 | 2000 | 2 | 248 | 479 |
| adult_seed_2_nrows_2000_nclasses_10_ncols_100_stratify_True | 15 | 9 | 2000 | 2 | 279 | 479 |
| adult_seed_3_nrows_2000_nclasses_10_ncols_100_stratify_True | 15 | 9 | 2000 | 2 | 254 | 479 |
| adult_seed_4_nrows_2000_nclasses_10_ncols_100_stratify_True | 15 | 9 | 2000 | 2 | 253 | 479 |
| rl_seed_0_nrows_2000_nclasses_10_ncols_100_stratify_True | 13 | 8 | 2000 | 2 | 0 | 1000 |
| wine_seed_0_nrows_2000_nclasses_10_ncols_100_stratify_True | 12 | 1 | 2000 | 2 | 0 | 1000 |
| Click_prediction_small_seed_0_nrows_2000_nclasses_10_ncols_100_stratify_True | 12 | 7 | 2000 | 2 | 0 | 337 |
| Amazon_employee_access_seed_0_nrows_2000_nclasses_10_ncols_100_stratify_True | 10 | 10 | 2000 | 2 | 0 | 116 |
| california_seed_0_nrows_2000_nclasses_10_ncols_100_stratify_True | 9 | 1 | 2000 | 2 | 0 | 1000 |
| sf-police-incidents_seed_0_nrows_2000_nclasses_10_ncols_100_stratify_True | 9 | 6 | 2000 | 2 | 0 | 243 |
| electricity_seed_0_nrows_2000_nclasses_10_ncols_100_stratify_True | 8 | 1 | 2000 | 2 | 0 | 1000 |
| airlines_seed_0_nrows_2000_nclasses_10_ncols_100_stratify_True | 8 | 5 | 2000 | 2 | 0 | 891 |
| mfeat-morphological | 7 | 1 | 2000 | 10 | 0 | 200 |
| jungle_chess_2pcs_raw_endgame_complete_seed_0_nrows_2000_nclasses_10_ncols_100_stratify_True | 7 | 1 | 2000 | 3 | 0 | 194 |
| phoneme_seed_0_nrows_2000_nclasses_10_ncols_100_stratify_True | 6 | 1 | 2000 | 2 | 0 | 1000 |
| wilt_seed_0_nrows_2000_nclasses_10_ncols_100_stratify_True | 6 | 1 | 2000 | 2 | 0 | 108 |
| steel-plates-fault | 34 | 1 | 1941 | 2 | 0 | 673 |
| steel-plates-fault_seed_0_nrows_2000_nclasses_10_ncols_100_stratify_True | 28 | 1 | 1941 | 7 | 0 | 55 |
| GAMETES_Epistasis_2-Way_20atts_0.1H_EDM-1_1 | 21 | 21 | 1600 | 2 | 0 | 800 |

*Table H.2.* Dataset statistics - Part 2

| Name | #Features | #Cat. Feat. | #Instances | #Classes | #NaNs | Minority Class Size |
|---|---|---|---|---|---|---|
| pc3 | 38 | 1 | 1563 | 2 | 0 | 160 |
| cmc | 10 | 8 | 1473 | 3 | 0 | 333 |
| cmc_seed_0_nrows_2000_nclasses_10_ncols_100_stratify_True | 10 | 8 | 1473 | 3 | 0 | 333 |
| ibm-employee-performance | 34 | 1 | 1470 | 2 | 0 | 226 |
| pc4 | 38 | 1 | 1458 | 2 | 0 | 178 |
| pc4_seed_0_nrows_2000_nclasses_10_ncols_100_stratify_True | 38 | 1 | 1458 | 2 | 0 | 178 |
| banknote-authentication | 5 | 1 | 1372 | 2 | 0 | 610 |
| analcatdata_halloffame | 17 | 2 | 1340 | 2 | 20 | 125 |
| mofn-3-7-10 | 11 | 11 | 1324 | 2 | 0 | 292 |
| socmob | 6 | 5 | 1156 | 2 | 0 | 256 |
| parity5_plus_5 | 11 | 11 | 1124 | 2 | 0 | 557 |
| PieChart3 | 38 | 1 | 1077 | 2 | 0 | 134 |
| qsar-biodeg | 42 | 1 | 1055 | 2 | 0 | 356 |
| qsar-biodeg_seed_0_nrows_2000_nclasses_10_ncols_100_stratify_True | 42 | 1 | 1055 | 2 | 0 | 356 |
| PizzaCutter3 | 38 | 1 | 1043 | 2 | 0 | 127 |
| rmftsa_sleepdata | 3 | 1 | 1024 | 4 | 0 | 94 |
| credit-g | 21 | 14 | 1000 | 2 | 0 | 300 |
| dummy | 7 | 1 | 1000 | 2 | 0 | 273 |
| xd6 | 10 | 10 | 973 | 2 | 0 | 322 |
| tokyo1 | 45 | 3 | 959 | 2 | 0 | 346 |
| tic-tac-toe | 10 | 10 | 958 | 2 | 0 | 332 |
| Tour-and-Travels-Customer-Churn-Prediction | 7 | 5 | 954 | 2 | 60 | 224 |
| stock | 10 | 1 | 950 | 2 | 0 | 462 |
| vehicle | 19 | 1 | 846 | 4 | 0 | 199 |
| vehicle_reproduced | 19 | 1 | 846 | 4 | 0 | 199 |
| analcatdata_authorship | 71 | 1 | 841 | 4 | 0 | 55 |
| analcatdata_dmft | 5 | 5 | 797 | 6 | 0 | 123 |
| diabetes | 9 | 1 | 768 | 2 | 0 | 268 |
| blood-transfusion-service-center | 5 | 1 | 748 | 2 | 0 | 178 |
| blood-transfusion-service-center_seed_0_nrows_2000_nclasses_10_ncols_100_stratify_True | 5 | 1 | 748 | 2 | 0 | 178 |
| doa_bwin_balanced | 14 | 3 | 708 | 2 | 0 | 354 |
| PieChart1 | 38 | 1 | 705 | 2 | 0 | 61 |
| breast-w | 10 | 1 | 699 | 2 | 16 | 241 |
| credit-approval | 16 | 10 | 690 | 2 | 67 | 307 |
| credit-approval_reproduced | 16 | 10 | 690 | 2 | 67 | 307 |
| Australian | 15 | 9 | 690 | 2 | 0 | 307 |
| Australian_seed_0_nrows_2000_nclasses_10_ncols_100_stratify_True | 15 | 9 | 690 | 2 | 0 | 307 |
| disclosure_x_bias | 4 | 1 | 662 | 2 | 0 | 317 |
| disclosure_x_tampered | 4 | 1 | 662 | 2 | 0 | 327 |
| disclosure_x_noise | 4 | 1 | 662 | 2 | 0 | 329 |
| disclosure_z | 4 | 1 | 662 | 2 | 0 | 314 |
| PizzaCutter1 | 38 | 1 | 661 | 2 | 0 | 52 |
| balance-scale | 5 | 1 | 625 | 3 | 0 | 49 |
| monks-problems-2 | 7 | 7 | 601 | 2 | 0 | 206 |
| synthetic_control | 61 | 1 | 600 | 6 | 0 | 100 |
| sensory | 12 | 12 | 576 | 2 | 0 | 239 |
| wdbc | 31 | 1 | 569 | 2 | 0 | 212 |
| arsenic-female-bladder | 5 | 2 | 559 | 2 | 0 | 80 |
| monks-problems-1 | 7 | 7 | 556 | 2 | 0 | 278 |
| monks-problems-3 | 7 | 7 | 554 | 2 | 0 | 266 |
| climate-model-simulation-crashes | 21 | 1 | 540 | 2 | 0 | 46 |
| doa_bwin | 14 | 3 | 530 | 2 | 0 | 176 |
| CPMP-2015-runtime-classification | 23 | 1 | 527 | 4 | 0 | 78 |
| kc2 | 22 | 1 | 522 | 2 | 0 | 107 |
| threeOf9 | 10 | 10 | 512 | 2 | 0 | 238 |
| rmftsa_ladata | 11 | 1 | 508 | 2 | 0 | 222 |
| boston_corrected | 21 | 4 | 506 | 2 | 0 | 223 |
| boston | 14 | 2 | 506 | 2 | 0 | 209 |
| collins | 23 | 3 | 500 | 2 | 0 | 80 |
| pm10 | 8 | 1 | 500 | 2 | 0 | 246 |
| no2 | 8 | 1 | 500 | 2 | 0 | 249 |
| LED-display-domain-7digit | 8 | 1 | 500 | 10 | 0 | 37 |
| irish | 6 | 4 | 500 | 2 | 32 | 222 |
| PopularKids | 11 | 5 | 478 | 3 | 0 | 90 |
| analcatdata_apnea2 | 4 | 3 | 475 | 2 | 0 | 64 |
| analcatdata_apnea1 | 4 | 3 | 475 | 2 | 0 | 61 |
| thoracic-surgery | 17 | 14 | 470 | 2 | 0 | 70 |
| analcatdata_vineyard | 4 | 2 | 468 | 2 | 0 | 208 |
| chscase_vine2 | 3 | 1 | 468 | 2 | 0 | 212 |
| sa-heart | 10 | 2 | 462 | 2 | 0 | 160 |
| analcatdata_apnea3 | 4 | 3 | 450 | 2 | 0 | 55 |
| wholesale-customers | 9 | 2 | 440 | 2 | 0 | 142 |
| mw1 | 38 | 1 | 403 | 2 | 0 | 31 |
| user-knowledge | 6 | 1 | 403 | 5 | 0 | 24 |

*Table H.3.* Dataset statistics - Part 3

| Name | #Features | #Cat. Feat. | #Instances | #Classes | #NaNs | Minority Class Size |
|---|---|---|---|---|---|---|
| chscase_census5 | 8 | 1 | 400 | 2 | 0 | 193 |
| chscase_census4 | 8 | 1 | 400 | 2 | 0 | 194 |
| chscase_census3 | 8 | 1 | 400 | 2 | 0 | 192 |
| chscase_census2 | 8 | 1 | 400 | 2 | 0 | 197 |
| chscase_census6 | 7 | 1 | 400 | 2 | 0 | 165 |
| analcatdata_germangss | 6 | 5 | 400 | 4 | 0 | 100 |
| calendarDOW | 33 | 21 | 399 | 5 | 0 | 44 |
| autoMpg | 8 | 4 | 398 | 2 | 6 | 189 |
| vinnie | 3 | 1 | 380 | 2 | 0 | 185 |
| jEdit_4.2_4.3 | 9 | 1 | 369 | 2 | 0 | 165 |
| dermatology | 35 | 34 | 366 | 6 | 8 | 20 |
| analcatdata_draft | 5 | 3 | 366 | 2 | 1 | 32 |
| analcatdata_birthday | 4 | 3 | 365 | 2 | 30 | 53 |
| ionosphere | 35 | 1 | 351 | 2 | 0 | 126 |
| SPECTF | 45 | 1 | 349 | 2 | 0 | 95 |
| penguins | 7 | 3 | 344 | 3 | 18 | 68 |
| CastMetal1 | 38 | 1 | 327 | 2 | 0 | 42 |
| visualizing_galaxy | 5 | 1 | 323 | 2 | 0 | 148 |
| plasma_retinol | 14 | 4 | 315 | 2 | 0 | 133 |
| solar-flare | 13 | 13 | 315 | 5 | 0 | 21 |
| diggle_table_a2 | 9 | 1 | 310 | 9 | 0 | 18 |
| vertebra-column | 7 | 1 | 310 | 3 | 0 | 60 |
| haberman | 4 | 2 | 306 | 2 | 0 | 81 |
| heart-c | 14 | 8 | 303 | 2 | 7 | 138 |
| cleveland | 14 | 8 | 303 | 2 | 6 | 139 |
| cholesterol | 14 | 8 | 303 | 2 | 6 | 137 |
| cleve | 14 | 9 | 303 | 2 | 0 | 138 |
| cleveland-nominal | 8 | 8 | 303 | 5 | 0 | 13 |
| CostaMadre1 | 38 | 1 | 296 | 2 | 0 | 38 |
| Heart_disease_prediction_20 | 14 | 1 | 296 | 2 | 0 | 137 |
| breast-cancer | 10 | 10 | 286 | 2 | 9 | 85 |
| breastTumor | 10 | 9 | 286 | 2 | 9 | 120 |
| analcatdata_broadwaymult | 8 | 5 | 285 | 7 | 27 | 21 |
| mu284 | 11 | 1 | 284 | 2 | 0 | 142 |
| DiabeticMellitus | 98 | 1 | 281 | 2 | 2 | 99 |
| breast-cancer-dropped-missing-attributes-values | 10 | 10 | 277 | 2 | 0 | 81 |
| jEdit_4.0_4.2 | 9 | 1 | 274 | 2 | 0 | 134 |
| heart-statlog | 14 | 1 | 270 | 2 | 0 | 120 |
| SPECT | 23 | 23 | 267 | 2 | 0 | 55 |
| Touch2 | 11 | 1 | 265 | 8 | 0 | 27 |
| analcatdata_lawsuit | 5 | 2 | 264 | 2 | 0 | 19 |
| rmftsa_ctoarrivals | 3 | 2 | 264 | 2 | 0 | 101 |
| MegaWatt1 | 38 | 1 | 253 | 2 | 0 | 27 |
| bodyfat | 15 | 1 | 252 | 2 | 0 | 124 |
| qualitative-bankruptcy | 7 | 7 | 250 | 2 | 0 | 107 |
| prnn_synth | 3 | 1 | 250 | 2 | 0 | 125 |
| conference_attendance | 7 | 7 | 246 | 2 | 0 | 31 |
| chatfield_4 | 13 | 1 | 235 | 2 | 0 | 93 |
| chscase_whale | 9 | 1 | 228 | 2 | 20 | 111 |
| lungcancer_GSE31210 | 24 | 3 | 226 | 2 | 0 | 35 |
| chscase_geyser1 | 3 | 1 | 222 | 2 | 0 | 88 |
| thyroid-new | 6 | 1 | 215 | 3 | 0 | 30 |
| glass | 10 | 1 | 214 | 6 | 0 | 9 |
| prnn_fglass | 10 | 1 | 214 | 2 | 0 | 76 |
| seeds | 8 | 1 | 210 | 3 | 0 | 70 |
| biomed | 9 | 2 | 209 | 2 | 15 | 75 |
| cpu | 8 | 2 | 209 | 2 | 0 | 53 |
| machine_cpu | 7 | 1 | 209 | 2 | 0 | 56 |
| sonar | 61 | 1 | 208 | 2 | 0 | 97 |
| regime_alimentaire | 20 | 17 | 202 | 2 | 17 | 41 |
| heart-long-beach | 14 | 1 | 200 | 5 | 0 | 10 |

*Table H.4.* Dataset statistics - Part 4

| Name | #Features | #Cat. Feat. | #Instances | #Classes | #NaNs | Minority Class Size |
|---|---|---|---|---|---|---|
| pwLinear | 11 | 1 | 200 | 2 | 0 | 97 |
| prnn_crabs | 8 | 2 | 200 | 2 | 0 | 100 |
| parkinsons | 23 | 1 | 195 | 2 | 0 | 48 |
| pharynx | 11 | 10 | 195 | 2 | 2 | 74 |
| KnuggetChase3 | 40 | 1 | 194 | 2 | 0 | 36 |
| wisconsin | 33 | 1 | 194 | 2 | 0 | 90 |
| lowbwt | 10 | 8 | 189 | 2 | 0 | 90 |
| triazines | 61 | 1 | 186 | 2 | 0 | 77 |
| chscase_funds | 3 | 1 | 185 | 2 | 0 | 87 |
| planning-relax | 13 | 1 | 182 | 2 | 0 | 52 |
| Smartphone-Based_Recognition_of_Human_Activities | 68 | 2 | 180 | 6 | 0 | 30 |
| backache | 32 | 27 | 180 | 2 | 0 | 25 |
| wine | 14 | 1 | 178 | 3 | 0 | 48 |
| servo | 5 | 5 | 167 | 2 | 0 | 38 |
| robot-failures-lp5 | 91 | 1 | 164 | 5 | 0 | 21 |
| analcatdata_wildcat | 6 | 3 | 163 | 2 | 0 | 47 |
| mc2 | 40 | 1 | 161 | 2 | 0 | 52 |
| corral | 7 | 7 | 160 | 2 | 0 | 70 |
| hayes-roth | 5 | 1 | 160 | 3 | 0 | 31 |
| auto_price | 16 | 2 | 159 | 2 | 0 | 54 |
| autoPrice | 16 | 1 | 159 | 2 | 0 | 54 |
| analcatdata_gsssexsurvey | 10 | 6 | 159 | 2 | 6 | 35 |
| TuningSVMs | 81 | 1 | 156 | 2 | 0 | 54 |
| grub-damage | 9 | 7 | 155 | 4 | 0 | 19 |
| teachingAssistant | 7 | 5 | 151 | 3 | 0 | 49 |
| tae | 6 | 3 | 151 | 3 | 0 | 49 |
| iris | 5 | 1 | 150 | 3 | 0 | 50 |
| iris-example | 5 | 1 | 150 | 3 | 0 | 50 |
| sleuth_case2002 | 7 | 5 | 147 | 2 | 0 | 69 |
| kc1-top5 | 95 | 1 | 145 | 2 | 0 | 8 |
| kc1-binary | 95 | 1 | 145 | 2 | 0 | 60 |
| newton_hema | 4 | 2 | 140 | 2 | 0 | 70 |
| veteran | 8 | 5 | 137 | 2 | 0 | 43 |
| analcatdata_boxing2 | 4 | 4 | 132 | 2 | 0 | 61 |
| analcatdata_seropositive | 4 | 2 | 132 | 2 | 0 | 46 |
| transplant | 4 | 1 | 131 | 2 | 0 | 48 |
| datatrieve | 9 | 1 | 130 | 2 | 0 | 11 |
| visualizing_livestock | 3 | 2 | 130 | 5 | 0 | 26 |
| humandevel | 2 | 1 | 130 | 2 | 0 | 65 |
| mux6 | 7 | 7 | 128 | 2 | 0 | 64 |
| MindCave2 | 40 | 1 | 125 | 2 | 0 | 44 |
| fruitfly | 5 | 3 | 125 | 2 | 0 | 49 |
| KungChi3 | 40 | 1 | 123 | 2 | 0 | 16 |
| heart-switzerland | 13 | 1 | 123 | 5 | 0 | 5 |
| ar1 | 30 | 1 | 121 | 2 | 0 | 9 |
| analcatdata_boxing1 | 4 | 4 | 120 | 2 | 0 | 42 |
| rabe_266 | 3 | 1 | 120 | 2 | 0 | 57 |
| robot-failures-lp4 | 91 | 1 | 117 | 3 | 0 | 21 |
| visualizing_environmental | 4 | 1 | 111 | 2 | 0 | 53 |
| cloud | 8 | 2 | 108 | 2 | 0 | 32 |
| analcatdata_michiganacc | 4 | 3 | 108 | 2 | 0 | 48 |
| ar4 | 30 | 1 | 107 | 2 | 0 | 20 |
| molecular-biology_promoters | 58 | 58 | 106 | 2 | 0 | 53 |
| breast-tissue | 10 | 1 | 106 | 6 | 0 | 14 |
| ar6 | 30 | 1 | 101 | 2 | 0 | 15 |
| zoo | 17 | 16 | 101 | 7 | 0 | 4 |
| fertility | 10 | 1 | 100 | 2 | 0 | 12 |
| analcatdata_creditscore | 7 | 4 | 100 | 2 | 0 | 27 |
| blogger | 6 | 6 | 100 | 2 | 0 | 32 |
| analcatdata_chlamydia | 4 | 4 | 100 | 2 | 0 | 19 |
| analcatdata_neavote | 3 | 2 | 100 | 2 | 0 | 7 |

# I. K-means Clustering for Dataset Selection

Table I.1 reports the extended results of our K-means clustering-based training set selection under different numbers of clusters $K \in \{3, 5, 10\}$, where a proportion ($P \in \{0.05, 0.10, 0.20\}$) of samples is drawn from each cluster. Across all settings, our method demonstrates stable performance regardless of $K$, confirming its robustness when applied with clustering-based selection.

*Table I.1.* **K-means-based training dataset selection.** Our method remains effective when training subsets are selected by clustering the data and sampling a percentage ($P$) of samples from each of $K$ clusters.

| $K$ | $P$ | Methods | w/o shift | Shift strength ($\beta$) | | | | | | |
|---|---|---|---|---|---|---|---|---|---|---|
| | | | | 0.0 | 0.1 | 0.5 | 1.0 | 2.0 | 5.0 | Avg. |
| 3 | 0.05 | TabPFN-v2 | 0.668 | 0.596 | 0.591 | 0.548 | 0.500 | 0.439 | 0.408 | 0.513 |
| | | DistPFN | 0.661 | 0.622 | 0.614 | 0.579 | 0.532 | 0.454 | 0.428 | 0.538 |
| | | DistPFN-T | 0.657 | **0.625** | **0.616** | **0.588** | **0.540** | **0.459** | **0.433** | **0.543** |
| | 0.10 | TabPFN-v2 | 0.699 | 0.626 | 0.632 | 0.570 | 0.528 | 0.465 | 0.424 | 0.541 |
| | | DistPFN | 0.692 | 0.641 | 0.653 | 0.620 | 0.564 | 0.498 | 0.450 | 0.573 |
| | | DistPFN-T | 0.687 | **0.643** | **0.657** | **0.628** | **0.569** | **0.504** | **0.454** | **0.584** |
| | 0.20 | TabPFN-v2 | 0.732 | 0.673 | 0.668 | 0.626 | 0.576 | 0.505 | 0.468 | 0.591 |
| | | DistPFN | 0.727 | 0.692 | 0.688 | 0.669 | 0.614 | 0.556 | 0.509 | 0.639 |
| | | DistPFN-T | 0.722 | **0.691** | **0.692** | **0.674** | **0.620** | **0.568** | **0.516** | **0.661** |
| 5 | 0.05 | TabPFN-v2 | 0.676 | 0.605 | 0.606 | 0.550 | 0.503 | 0.459 | 0.429 | 0.529 |
| | | DistPFN | 0.673 | 0.628 | 0.630 | 0.587 | 0.534 | 0.487 | 0.453 | 0.561 |
| | | DistPFN-T | 0.672 | **0.629** | **0.634** | **0.594** | **0.539** | **0.493** | **0.460** | **0.565** |
| | 0.10 | TabPFN-v2 | 0.699 | 0.631 | 0.644 | 0.586 | 0.540 | 0.485 | 0.446 | 0.569 |
| | | DistPFN | 0.696 | 0.654 | 0.665 | 0.624 | 0.583 | 0.528 | 0.475 | 0.609 |
| | | DistPFN-T | 0.693 | **0.655** | **0.670** | **0.630** | **0.591** | **0.538** | **0.483** | **0.620** |
| | 0.20 | TabPFN-v2 | 0.732 | 0.670 | 0.679 | 0.628 | 0.582 | 0.523 | 0.481 | 0.618 |
| | | DistPFN | 0.736 | 0.687 | 0.697 | 0.662 | 0.625 | 0.576 | 0.521 | 0.645 |
| | | DistPFN-T | 0.731 | **0.690** | **0.698** | **0.670** | **0.631** | **0.584** | **0.531** | **0.667** |
| 10 | 0.05 | TabPFN-v2 | 0.708 | 0.644 | 0.639 | 0.591 | 0.547 | 0.505 | 0.460 | 0.554 |
| | | DistPFN | 0.706 | 0.659 | 0.662 | 0.627 | 0.586 | 0.548 | 0.493 | 0.589 |
| | | DistPFN-T | 0.701 | **0.662** | **0.667** | **0.640** | **0.601** | **0.562** | **0.504** | **0.605** |
| | 0.10 | TabPFN-v2 | 0.727 | 0.663 | 0.664 | 0.617 | 0.582 | 0.534 | 0.481 | 0.620 |
| | | DistPFN | 0.723 | 0.679 | 0.685 | 0.650 | 0.618 | 0.577 | 0.515 | 0.642 |
| | | DistPFN-T | 0.718 | **0.685** | **0.691** | **0.656** | **0.629** | **0.588** | **0.527** | **0.652** |
| | 0.20 | TabPFN-v2 | 0.749 | 0.697 | 0.689 | 0.651 | 0.619 | 0.561 | 0.510 | 0.638 |
| | | DistPFN | 0.749 | 0.713 | 0.711 | 0.682 | 0.663 | 0.610 | 0.553 | 0.676 |
| | | DistPFN-T | 0.748 | **0.716** | **0.715** | **0.689** | **0.670** | **0.620** | **0.563** | **0.688** |

## J. Application to LoCalPFN

Table J.1 provides the full results for LoCalPFN under different values of $k$ across six $\beta$ values. The results confirm that our methods yield consistent improvements regardless of the choice of $k$, demonstrating robustness of the approach.

*Table J.1.* **Application to LoCalPFN.** DistPFN and DistPFN-T applied to LoCalPFN show consistent improvements across varying numbers of neighbors ($k$).

| $k$ | Methods | Shift strength ($\beta$) | | | | | | Avg. |
| --- | --- | --- | --- | --- | --- | --- | --- | --- |
| | | 0.0 | 0.1 | 0.5 | 1.0 | 2.0 | 5.0 | |
| 3 | LoCalPFN | 0.789 | 0.787 | 0.774 | 0.758 | 0.711 | 0.679 | 0.750 |
| | + DistPFN | 0.794 | 0.794 | 0.792 | 0.786 | 0.772 | 0.752 | 0.782 |
| | + DistPFN-T | **0.794** | **0.794** | **0.794** | **0.790** | **0.779** | **0.759** | **0.785** |
| 5 | LoCalPFN | 0.792 | 0.791 | 0.785 | 0.775 | 0.744 | 0.714 | 0.767 |
| | + DistPFN | 0.794 | 0.795 | 0.793 | 0.790 | 0.777 | 0.766 | 0.786 |
| | + DistPFN-T | **0.795** | **0.796** | **0.795** | **0.794** | **0.784** | **0.770** | **0.789** |
| 10 | LoCalPFN | 0.794 | 0.792 | 0.786 | 0.778 | 0.752 | 0.720 | 0.770 |
| | + DistPFN | 0.796 | 0.795 | 0.793 | 0.791 | 0.779 | 0.768 | 0.787 |
| | + DistPFN-T | **0.797** | **0.797** | **0.796** | **0.794** | **0.785** | **0.774** | **0.789** |
| 20 | LoCalPFN | 0.794 | 0.793 | 0.788 | 0.778 | 0.753 | 0.719 | 0.771 |
| | + DistPFN | 0.797 | 0.796 | 0.794 | 0.790 | 0.782 | 0.770 | 0.788 |
| | + DistPFN-T | **0.798** | **0.797** | **0.796** | **0.794** | **0.787** | **0.776** | **0.791** |

# K. Comparison with Methods for Label Shift Correction

To demonstrate the effectiveness of our approach, we compare it with classical methods for handling label shift by rescaling classifier outputs, which typically require estimating the test distribution: EM-based Estimation (EME) (Saerens et al., 2002) and Black-box Estimation (BBE) (Lipton et al., 2018). Table K.1 presents the results, showing that our method is effective without requiring estimation of the test prior.

*Figure K.1.* Comparison with other label shift methods.

| Methods | w/o shift | Shift strength ($\beta$) | | | | | | |
|---|---|---|---|---|---|---|---|---|
| | | 0.0 | 0.1 | 0.5 | 1.0 | 2.0 | 5.0 | Avg. |
| LoCalPFN | **0.816** | 0.794 | 0.793 | 0.788 | 0.778 | 0.753 | 0.719 | 0.771 |
| + EME | 0.801 | 0.792 | 0.790 | 0.786 | 0.785 | 0.778 | 0.769 | 0.783 |
| + BBE | 0.805 | **0.798** | 0.795 | 0.792 | 0.789 | 0.782 | 0.770 | 0.787 |
| + DistPFN | **0.816** | 0.797 | 0.796 | 0.794 | 0.790 | 0.782 | 0.770 | 0.788 |
| + DistPFN-T | **0.816** | **0.798** | **0.797** | **0.796** | **0.794** | **0.787** | **0.776** | **0.791** |
| TabICL | **0.806** | 0.783 | 0.781 | 0.770 | 0.747 | 0.704 | 0.664 | 0.742 |
| + EME | 0.798 | 0.776 | 0.776 | 0.770 | 0.769 | 0.761 | 0.747 | 0.766 |
| + BBE | 0.802 | 0.783 | 0.785 | 0.780 | 0.774 | 0.754 | 0.734 | 0.768 |
| + DistPFN | **0.806** | 0.786 | 0.786 | 0.781 | 0.776 | 0.763 | 0.746 | 0.773 |
| + DistPFN-T | **0.806** | **0.786** | **0.786** | **0.783** | **0.780** | **0.771** | **0.755** | **0.777** |
| TabPFN-v2 | **0.818** | 0.797 | 0.796 | 0.790 | 0.782 | 0.759 | 0.727 | 0.775 |
| + EME | 0.801 | 0.793 | 0.793 | 0.790 | 0.787 | 0.783 | 0.768 | 0.786 |
| + BBE | 0.805 | **0.799** | 0.797 | **0.797** | 0.791 | 0.783 | 0.768 | 0.789 |
| + DistPFN | **0.818** | **0.799** | 0.797 | 0.795 | 0.791 | 0.783 | 0.769 | 0.789 |
| + DistPFN-T | **0.818** | **0.799** | **0.798** | **0.797** | **0.796** | **0.789** | **0.775** | **0.792** |

# L. Predicted Distribution of Single vs. Multiple Instances

As TabPFN produces identical predictions whether test instances are evaluated individually or in batches, DistPFN and DistPFN-T can adjust based on either 1) the prediction of a *single* instance or 2) the average prediction across *multiple* instances. As shown in Table L.1, both choices consistently improve TabPFN-v2 (Hollmann et al., 2025), averaged across six $\beta$s for *w/ shift*, demonstrating robustness to the choice of distribution source.

*Table L.1.* **Predicted distributions: Single vs. Multiple.** The proposed methods consistently improves TabPFN-v2 regardless of whether the adjustment is based on single or aggregated distribution.

| | Pred. distn. | w/o shift | Shift strength ($\beta$) | | | | | | |
|---|---|---|---|---|---|---|---|---|---|
| | | | 0.0 | 0.1 | 0.5 | 1.0 | 2.0 | 5.0 | Avg. |
| TabPFN-v2 | - | 0.818 | 0.797 | 0.796 | 0.790 | 0.782 | 0.759 | 0.727 | 0.775 |
| + DistPFN | Single | **0.818** | 0.797 | 0.796 | **0.795** | **0.793** | **0.784** | **0.770** | **0.789** |
| | Multiple | **0.818** | **0.799** | **0.797** | **0.795** | 0.791 | 0.783 | **0.770** | **0.789** |
| + DistPFN-T | Single | **0.818** | 0.797 | 0.797 | 0.796 | 0.795 | 0.788 | 0.773 | 0.791 |
| | Multiple | **0.818** | **0.799** | **0.798** | **0.797** | **0.796** | **0.789** | **0.775** | **0.792** |

# M. Results on OpenML-CC18 Benchmark

To verify that our findings are not an artifact of dataset selection, we additionally report results on the OpenML-CC18 benchmark (Bischl et al., 2021), a widely used curated subset of 25 datasets, averaged over 5 seeds.

*Table M.1.* Results on OpenML-CC18 subset (25 datasets).

| $\beta$ | 0.0 | 0.1 | 0.5 | 1.0 | 2.0 | 5.0 |
|---|---|---|---|---|---|---|
| TabPFN-v2 | .845 | .845 | .839 | .832 | .796 | .731 |
| + DistPFN | .845 | .845 | .840 | .837 | .827 | .808 |
| + DistPFN-T | **.846** | **.846** | **.842** | **.839** | **.831** | **.816** |

The conclusions are consistent with the full evaluation on 253 datasets, confirming robustness to benchmark selection.

# N. Results on Extremely Imbalanced Datasets

We select the top-20 most imbalanced datasets from 253 OpenML datasets, where the balance ratio (minority class size / expected uniform class size) ranges from 0.108 to 0.213, and evaluate DistPFN-T across all $\beta$ values, averaged over 5 seeds.

*Table N.1.* DistPFN-T on the top-20 most imbalanced datasets.

| Method | $\beta$=0.0 | $\beta$=0.5 | $\beta$=1.0 | $\beta$=2.0 | $\beta$=5.0 |
|---|---|---|---|---|---|
| TabPFN-v2 | **0.866** | **0.859** | 0.858 | 0.824 | 0.829 |
| + DistPFN | **0.866** | **0.859** | 0.858 | 0.851 | 0.846 |
| + DistPFN-T | **0.866** | **0.859** | **0.859** | **0.855** | **0.850** |

Even on extremely imbalanced datasets, DistPFN-T's improvement grows as label shift increases, confirming effectiveness precisely when the bias is most severe.

