# OpenReview forum: "Mitigating Label Shift in Tabular In-Context Learning via Test-Time Posterior Adjustment"
_ICML.cc/2026/Conference — ICML 2026 regular_

### Official Review · Reviewer_B5fU · 2026-02-15

**Soundness:** 3
**Presentation:** 2
**Significance:** 3
**Originality:** 2
**Overall Recommendation:** 4
**Confidence:** 3

**Summary:**

The paper tackles the problem of label shift for tabular foundation models (TFMs). They propose simple training-free heuristics to push the distribution over classes that the TFM outputs closer to the distribution over the labels in the training-set. In detailed experiments they confirm that their method yields consistent benefits in the case where there is a label shift in the data.

**Compliance With Llm Reviewing Policy:**

Affirmed.

**Final Justification:**

The authors successfully answered my questions, and addressed my concerns in the rebuttal.


I adjusted my score accordingly from 3 to 4.

**Key Questions For Authors:**

Why can't the method the method be applied to any method that outputs class probabilities? Why the focus on TFMs?
Why do the authors benchmark on 250 OpenML datasets, and not an established benchmark?
Why is the method specific to TFMs?

**Limitations:**

yes

**Strengths And Weaknesses:**

Presentation

The tackled problem becomes very clear in the introduction. However, the usage of the term "prior" in the paper is confusing to me. For instance here: "by reweighting class probabilities based on the ratio between
the posterior and the class distribution of the training dataset (i.e., prior)." Why is the prior the same as the training distribution of labels? I understand what is meant, but this usage of the term "prior" is new to me.

Also, in Figure 1/Table 1, please at least mention on which dataset(s) are used.

I appreciate when a paper contains a lot of plots and figures, but maybe it is a bit too much for this paper. Some plots could certainly go to the Appendix without the paper loosing anything.

In terms of experiments, the section is very dense in the sense that a lot of information is conveyed in a small amount of space without a clear narrative. Motivating why certain experiments are conducted could help the reader.

Also, the impact statement is missing.

Soundness:
Conceptually, the proposed method makes sense, but there is no deeper justification besides basic arguments in the appendix provided.

The experiments are very detailed and the results seem to be sound within the scope of the experiments.

In terms of the benchmark, the selection of datasets is problematic; there are important reasons why established benchmarks don't use ALL datasets from OpenML (https://proceedings.neurips.cc/paper_files/paper/2022/file/0378c7692da36807bdec87ab043cdadc-Paper-Datasets_and_Benchmarks.pdf, https://arxiv.org/abs/2506.16791).

Also, the choice of baselines, i.e. TabM, FT-Transfromer, TabularRNN etc. is weird---why are there no tree-based methods in Table 4, for instance?

Also, no error bars are ever reported.

Significance:

While I appreciate that the proposed method is simple, I am concerned that it is not really a meaningful advancement, but more a "quick-fix" for a specific shortcoming of current TFMs; i.e. I am not convinced that the problem that is tackled is truly fundamental or super meaningful since more comprehensive pre-training, also under label shift, might very easily resolve the issue discussed here.
However, until then this method might be practically useful.

Originality:

It is certainly meaningful that the drawback of current TFMs under label shift is discussed---I was not aware of that. The solution is perhaps not very original, but straightforward and simple, but otherwise not very deep. Maybe it would be helpful to put the discussion on conceptual justifications from the appendix to the main body of the paper.

Summary:

My main concern is the soundness of the experiments. Since the proposed method does not have meaningful theoretical underpinnings, empirical results are key. However,  benchmark-dataset selection is questionable and the baselines non-standard. Furthermore, I do not understand why the method is only evaluated for TFMs.

NIT:
- It is "TuneTables" and not "TuneTable"

---

> ### Author Rebuttal · Authors · 2026-03-27
>
> We thank the reviewer for the detailed feedback. We address each point below.
>
> &nbsp;
>
> ---
>
> ## [Presentation]
>
> ### W1: Term "prior" is confusing
>
> The term follows the label shift correction literature, where $P_{train}(Y)$ is referred to as the "source prior."
> We acknowledge that this may cause confusion, and will clarify this in the final version.
>
> &nbsp;
>
> ### W2: Figure 1/Table 1 should mention the dataset
>
> Thank you for pointing it out. We will specify the dataset (*CostaMadre1*) in the final version.
>
> &nbsp;
>
> ### W3: Too many plots/figures
>
> Our paper includes numerous figures/tables to provide comprehensive analysis with multiple dimensions.
> Following the reviewer's suggestion, we will move supplementary figures to the **Appendix** and keep only core results in the main body.
>
> &nbsp;
>
> ### W4: Experiments section too dense
>
> We originally aimed for a compact presentation given the breadth of experiments.
> Following the reviewer's feedback, we will add motivating sentences at the beginning of each subsection to clarify the purpose of each experiment.
>
> &nbsp;
>
> ### W5: Impact statement missing
>
> We will add it in the final version.
>
> &nbsp;
>
> &nbsp;
>
> ---
>
> ## [Soundness]
>
> ### W6: No deeper theoretical justification
>
> Thank you for this feedback. Our primary focus is on the **empirical effectiveness** of DistPFN as a **plug-in** method applicable to **any TFM without modification**. Developing a deeper theoretical framework is a interesting work that falls ***outside the scope of our work***, which we leave as a future work. We are also open to moving the conceptual justification in the main body.
>
> &nbsp;
>
> ### W7: Benchmark dataset selection
>
> We chose 253 OpenML datasets following TabPFN-v2 to ensure **wide coverage of diverse class distributions**, as curated subsets may exclude extreme class imbalance cases.
>
> That said, to address the reviewer's concern, we additionally report results on the **OpenML-CC18 benchmark** [1], a widely used curated subset, averaged over 5 seeds:
>
> **Table R1. Results on OpenML-CC18 subset (25 datasets).**
>
> |β|0.0|0.1|0.5|1.0|2.0|5.0|
> |-|-|-|-|-|-|-|
> |TabPFN-v2|.845|.845|.839|.832|.796|.731|
> |+ DistPFN|.845|.845|.840|.837|.827|.808|
> |+ DistPFN-T|**.846**|**.846**|**.842**|**.839**|**.831**|**.816**|
>
> Conclusions are **consistent** with the full evaluation, confirming our findings are not an artifact of dataset selection.
>
> - [1] Bischl et al., "OpenML Benchmarking Suites," NeurIPS D&B, 2021.
>
> &nbsp;
>
> ### W8: No tree-based methods in Table 4
>
> - Tree-based baselines are included in **Appendix G** due to space limit (noted at the end of **Section 5.2**).
> - Additionally, we also applied DistPFN to these models — due to the word limit, please refer to **Reviewer wwaB [W1]** for full results showing **meaningful improvements even for tree-based models**.
>
> &nbsp;
>
> ### W9: No error bars
>
> We would like to clarify that **Table 4** in the paper already reports standard deviation across 5 seeds.  We will add error bands in the final version.
>
> &nbsp;
>
> &nbsp;
>
> ---
>
> ## [Significance]
>
> ### W10: Could more comprehensive pretraining resolve the issue?
>
> We respectfully offer several counterpoints:
>
> - 1. **Pretraining cannot cover all shifts**: Real-world class distributions are infinitely diverse and change over time, and it is impractical to include every possible label shift scenario during pretraining.
> - 2. **Complementary, not competing**: DistPFN operates *on top of* pretrained models. Even if future pretraining strategies become more robust to label shift, DistPFN can still provide additional gains — the two approaches are not mutually exclusive.
> - 3. **Practical necessity**: In real deployment, class distributions shift over time, and test-time adjustment is often the *only* viable option when model retraining is not feasible.
>
> &nbsp;
>
> &nbsp;
>
> ---
>
> ## [Key Questions]
>
> ### Q1–Q4: Why focus on TFMs? Why 250 OpenML datasets?
>
> ***Why focus on TFMs?***
> - As discussed in **Table 2**, TFMs are **Explicit models** that *directly reference training data during inference*, meaning the training prior explicitly influences the output predictions. We originally designed DistPFN to target this structural property.
> - That said, DistPFN is **technically applicable** to any classifier that outputs class probabilities, including **Implicit models** where the prior is absorbed into parameters. Following the reviewer's suggestion, we conducted additional experiments on tree-based models and found that **DistPFN yields improvements even for these models** (**Reviewer wwaB [W1]**).
>
> ***Why 250 OpenML datasets?***
> - Label shift research requires evaluating across diverse class distributions, and a curated benchmark may exclude the highly imbalanced datasets. As shown in **Table R1**, results on OpenML-CC18 subset are consistent with our full evaluation, confirming that our conclusions are robust to benchmark selection.
>
>
> &nbsp;
>
> &nbsp;
>
> **Please let us know if there are any remaining issues you'd like to discuss!**

---

> > ### Author Rebuttal · Reviewer_B5fU · 2026-04-01
> >
> > Thank you for your response. I believe that
> >
> > (a) the authors can improve the presentation of the paper with their suggested changes for the camera-ready version.
> > (b) that the additional empirical findings (on CC18 and for the tree-based ablation) meaningfully improve the soundness of the paper.
> >
> > I adjusted my score accordingly.

---

> > > ### Author Response · Authors · 2026-04-01
> > >
> > > We sincerely thank the reviewer for the feedback and for recognizing the improvements, as well as for adjusting the score accordingly.

---

### Official Review · Reviewer_QgQF · 2026-03-09

**Soundness:** 3
**Presentation:** 3
**Significance:** 2
**Originality:** 2
**Overall Recommendation:** 4
**Confidence:** 5

**Summary:**

To address the issue of majority class bias in tabular-based models like TabPFN due to the susceptibility of training set class distribution to label shifts, the authors propose a test-time posterior probability adjustment method called DistPFN and its adaptive temperature scaling variant. By reweighting the output probabilities during the inference phase based on the ratio of training priors to model predictions, the robustness of the model to class distribution shifts is significantly enhanced without the need for retraining or architectural modifications.

**Compliance With Llm Reviewing Policy:**

Affirmed.

**Final Justification:**

Thanks to the author for resolving all my concerns, I remain my evaluation unchanged.

**Key Questions For Authors:**

Q1: Regarding DistPFN-T, why was Cross-Entropy (CE) selected for temperature scaling instead of KL divergence or other distance metrics?
Q2: Since the method is technically applicable to traditional tree-based models (e.g., XGBoost), have the authors conducted experiments on these baselines to verify if the performance gains generalize?

**Limitations:**

yes

**Strengths And Weaknesses:**

Please provide a thorough assessment of the strengths and weaknesses of the paper, touching on each of the following dimensions: soundness, presentation, significance, and originality. We encourage you to be open-minded about the potential strengths and broad definitions of significance and originality. For example, originality may arise from creative combinations of existing ideas, application to a real-world use case, or removing restrictive assumptions from prior theoretical results. We provide detailed guidelines below on each dimension:
# Strengths
S1 ：Large-scale experimental validation: The authors benchmarked on over 250 OpenML datasets, a scale of experimentation that is robust and statistically convincing for tabular data research.

S2：Plug-and-play usability: The proposed DistPFN requires no modification to the model architecture or any parameter updates, which is highly attractive for computationally expensive Transformer base models.

S3：Insightful observation of the limitations of tabular FMs: Identifying the majority class bias of TabPFN under label offsets is a valuable insight, providing insights into the behavior of ICL in non-linguistic tasks.

# Weaknesses
W1:  Using priors for posterior adjustment is standard in long-tail learning and domain adaptation. Approximating test distributions via model means is common in semi-supervised learning and self-training [1, 2, 3]. The authors fail to justify the uniqueness of this method compared to these established works.

W2: DistPFN uses model predictions as a proxy for test priors. This results in circular reasoning if initial predictions are severely biased. The paper lacks a failure mode analysis for cases where the foundation model produces fundamentally unreliable initial results.

W3: The cross-entropy based temperature scaling in DistPFN-T is heuristic. While empirical gains are observed, there is no mathematical proof that cross-entropy is the optimal metric for adjustment strength. This design choice lacks theoretical grounding.

## Reference
[1] Saerens M, Latinne P, Decaestecker C. Adjusting the outputs of a classifier to new a priori probabilities: a simple procedure[J]. Neural computation, 2002, 14(1): 21-41.

[2] Menon A K, Jayasumana S, Rawat A S, et al. Long-tail learning via logit adjustment[J]. arXiv preprint arXiv:2007.07314, 2020.

[3] Ren J, Yu C, Ma X, et al. Balanced meta-softmax for long-tailed visual recognition[J]. Advances in neural information processing systems, 2020, 33: 4175-4186.

---

> ### Author Rebuttal · Authors · 2026-03-27
>
> We thank the reviewer for the detailed feedback. We address each point below.
>
> &nbsp;
>
> &nbsp;
>
> ---
>
> ## W1: Prior adjustment is standard in long-tail learning; uniqueness not justified
>
> We appreciate this comment and clarify the key differences between DistPFN and the cited works:
>
> ||[1]|[2]|[3]|**DistPFN (Ours)**|
> |-|-|-|-|-|
> |*When applied*|Post-hoc|Training|Training|**Post-hoc**|
> |*Test prior estimation*|✅|❌|❌|❌|
> |*Model retraining*|❌|✅|✅|❌|
>
> The key distinctions are:
> - 1. **vs. [1]**: Both are post-hoc methods, but [1] requires **iterative EM estimation of the test prior**, which involves multiple passes over the test data until convergence, while DistPFN requires **no test prior estimation**.
> - 2. **vs. [2] / [3]**: These methods adjust logits/softmax **during training**, requiring model retraining. DistPFN operates on **pre-trained TFMs without any modification**, making it applicable to any TFMs.
>
> We have also empirically compared against other methods (EME [1], BBE) in **Table 12**, where DistPFN-T outperforms both.
>
> &nbsp;
>
> &nbsp;
>
> ---
>
> ## W2: Circular reasoning if initial predictions are severely biased
>
> The key insight is that DistPFN performs a **partial** correction, not a full one. Even when TabPFN's predictions are biased toward the training prior, the predicted posterior is unlikely to fully collapse to the prior, as it still reflects learned patterns from the data. Therefore, the adjustment *partially* removes the prior's influence, acting as a **conservative correction** rather than a circular one. The Oracle comparison (**Table 8**) confirms this: the gap between DistPFN and the oracle is small, indicating **the proxy is effective despite being imperfect**.
>
> &nbsp;
>
> Empirically, we conducted a failure mode analysis across all 253 datasets (5 seeds):
>
> **Table R1. Accuracy improvement by base model accuracy quartile (β=5.0).**
>
> |Base Acc. Quartile|Mean Δ Acc. (DistPFN-T)|
> |-|-|
> |Q1|+.124|
> |Q2|+.064|
> |Q3|+.012|
> |Q4|+.002|
>
> Even when the base model produces low-accuracy predictions (Q1), DistPFN-T maintains an improvement, suggesting that circular reasoning is unlikely to be a failure mode.
>
> &nbsp;
>
> &nbsp;
>
> ---
>
> ## W3: Cross-entropy based temperature scaling is heuristic; lacks theoretical grounding
>
> We would like to clarify that the core idea of DistPFN-T is **NOT** that "cross-entropy is the optimal metric," but rather that ***the adjustment strength should be proportional to the discrepancy between the predicted posterior and the training prior***. The principle of using distributional discrepancy as a temperature is the main contribution; the specific choice of distance metric is secondary.
>
> &nbsp;
>
> That said, we chose CE as the default for two reasons:
>
> **(1) Scale suitability:**
> In DistPFN-T, the metric value is directly used as τ. This means the metric must produce values in a suitable range — if τ ≈ 0, the softmax becomes a hard argmax and predictions collapse. We computed the actual τ values across 253 datasets at β=2.0:
>
> **Table R3-a. τ value statistics.**
>
> |Metric|Median|Mean|
> |-|-|-|
> |CE|1.004|1.339|
> |KL|0.162|0.497|
> |JS|0.040|0.090|
> |L2|0.334|0.396|
>
> CE produces τ values with a median of ~1.0, yielding appropriate softmax smoothing, compared to other metrics.
>
> &nbsp;
>
> **(2) Mathematical naturalness:**
> CE is composed of two terms, which simultaneously capture:
> - **KL(p̂ ∥ p_train)**: the actual divergence between predicted posterior and training prior
> - **H(p̂)**: the entropy (uncertainty) of the prediction itself
>
> The H(p̂) term acts as a **natural lower bound**, which helps prevent τ from collapsing to zero even when the two distributions are similar. We believe this is a key reason why CE tends to maintain a suitable scale without requiring manual rescaling.
>
> &nbsp;
>
> **Metric ablation:**
> - Following the reviewer's feedback, we conducted an ablation study using four metrics for τ without rescaling, averaged over 5 seeds:
>
> **Table R3-b. τ metric ablation (TabPFN-v2, avg over 253 datasets).**
>
> |β|0.0|0.5|1.0|2.0|5.0|
> |-|-|-|-|-|-|
> |CE (ours)|.802|.800|.798|.790|.782|
> |L2 dist|.788|.785|.778|.774|.761|
> |KL div|.649|.683|.690|.694|.696|
> |JS div|.614|.622|.629|.636|.641|
>
> **CE achieves the best performance**, and the performance gap appears to be largely driven by **scale suitability** rather than the divergence measure itself.
> These results support CE as a reasonable default. We will add this analysis in the final version.
>
> &nbsp;
>
> &nbsp;
>
> ---
>
> ## Q1: Why  CE  instead of other distance metrics?
>
> Please see **[W3]** above.
>
> &nbsp;
>
> &nbsp;
>
> ---
>
> ## Q2: Experiments on tree-based baselines
>
> - Tree-based baselines are included in **Appendix G** due to space limit (noted at the end of **Section 5.2**).
> - We also applied DistPFN to these models — due to the word limit, please refer to **Reviewer wwaB [W1]** for full results showing **meaningful improvements even for tree-based models**.
>
> &nbsp;
>
> &nbsp;
>
> **Please let us know if there are any remaining issues you'd like to discuss!**

---

> > ### Author Rebuttal · Reviewer_QgQF · 2026-04-02
> >
> > Thanks to the author for resolving all my concerns.

---

> > > ### Author Response · Authors · 2026-04-03
> > >
> > > We sincerely thank the reviewer for acknowledging that all concerns have been addressed.

---

### Official Review · Reviewer_wwaB · 2026-03-11

**Soundness:** 4
**Presentation:** 4
**Significance:** 4
**Originality:** 3
**Overall Recommendation:** 5
**Confidence:** 3

**Summary:**

This paper focuses on a core concept: improving the robustness of foundation models for tabular data—such as TabPFN—when confronted with Label Shift. The authors observe that TabPFN is prone to overfitting to the majority classes within the training data, resulting in severe performance degradation in test scenarios where the class distribution has shifted. To address this, the paper proposes DistPFN, the first post-hoc adjustment method designed specifically for tabular foundation models. This method mitigates majority-class bias by rescaling predicted class probabilities through division by the training priors. Furthermore, the authors introduce temperature scaling to dynamically control the intensity of this adjustment based on the discrepancy between the priors and the posteriors. Experimental results across more than 250 OpenML datasets demonstrate that this method significantly enhances classification accuracy in environments with Label Shift—without altering the model architecture or incurring additional training costs—while simultaneously maintaining robustness under standard settings.

**Compliance With Llm Reviewing Policy:**

Affirmed.

**Final Justification:**

Overall, I think this work remains at an acceptable level, and the author's responses have also been quite clear, resolving my concerns.

**Key Questions For Authors:**

1.Could the author please comment on how this method further addresses distribution shifts within the features themselves, and the specific mechanisms through which it takes effect?

**Limitations:**

yes

**Strengths And Weaknesses:**

The authors address a crucial issue: uneven class distribution, a common problem in practical applications of tabular data models. DistPFN employs a similar approach to classic label shift correction, but cleverly leverages the tabular model's ability to directly reference training data as context, enabling plug-and-play adjustments without explicit prior estimation of the test set. Experiments cover over 250 datasets, encompassing not only synthetic label shift scenarios but also real-world tasks like TableShift, demonstrating significant statistical value. Furthermore, since the adjustment occurs only at the probability output layer during inference, this method adds almost no inference latency.

This approach heavily relies on context learning models that can directly access training samples during inference. Whether it's also an effective application path for traditional models like random forests or ordinary neural networks remains to be seen. Additionally, while the paper uses dynamic scaling to stabilize the adjustment strength, further verification is needed to determine its effectiveness in extreme cases of significant data bias.

---

> ### Author Rebuttal · Authors · 2026-03-26
>
> We sincerely thank the reviewer for the positive assessment and for recognizing the significance of our work.
>
> We address the question and concerns below.
>
> &nbsp;
>
> ---
>
> ### [W1] Applicability to traditional models
>
> > *"Whether it's also an effective application path for ***traditional models*** like random forests ..."*
>
> As discussed in **Table 2** of our paper, DistPFN is specifically designed for models that ***explicitly reference training data during inference*** (e.g., TabPFN, TabICL, LoCalPFN). In these models, the **training prior directly influences predictions** through attention over the training context, making output-level adjustment effective.
>
> Traditional models (Random Forest, XGBoost, etc.) encode the training prior ***implicitly*** in their model parameters during training. Nonetheless, since DistPFN can technically be applied to any model that outputs class probabilities, we followed the reviewer's suggestion and conducted **additional experiments on tree-based models** (XGBoost, RandomForest, LightGBM, each with 100 estimators and default hyperparameters):
>
> **Table R1. DistPFN applied to tree-based models (averaged over 253 datasets, 5 seeds).**
>
> | Method | β=0.0 | β=0.1 | β=0.5 | β=1.0 | β=2.0 | β=5.0 |
> |-|-|-|-|-|-|-|
> | **XGBoost** | 0.763 | 0.759 | 0.743 | 0.715 | 0.664 | 0.623 |
> | + DistPFN | **0.770** | **0.768** | **0.758** | 0.742 | 0.703 | 0.672 |
> | + DistPFN-T | 0.768 | **0.768** | **0.758** | **0.743** | **0.710** | **0.679** |
> |-|-|-|
> | **RandomForest** | 0.769 | 0.767 | 0.750 | 0.720 | 0.664 | 0.617 |
> | + DistPFN | **0.776** | **0.776** | 0.771 | 0.764 | 0.742 | 0.717 |
> | + DistPFN-T | **0.776** | **0.776** | **0.772** | **0.768** | **0.752** | **0.730** |
> |-|-|-|
> | **LightGBM** | 0.758 | 0.753 | 0.734 | 0.705 | 0.657 | 0.618 |
> | + DistPFN | **0.764** | **0.761** | **0.748** | 0.727 | 0.682 | 0.645 |
> | + DistPFN-T | **0.764** | **0.761** | **0.748** | **0.728** | **0.687** | **0.650** |
>
> As shown in the table above, DistPFN-T yields meaningful improvements even for tree-based models, with gains consistently increasing as β grows, indicating stronger effectiveness under larger label shift. While DistPFN was originally designed for Explicit models (**Table 2**), the output-level adjustment proves broadly effective as a **general-purpose post-hoc correction** across model families.
>
>
> &nbsp;
>
> &nbsp;
>
> ---
>
> ### [W2] Effectiveness in extreme data bias
>
> > *"Additionally, ... further verification is needed to determine its effectiveness in ***extreme cases of significant data bias***."*
>
> To directly address this concern, we selected the **top-20 most imbalanced datasets** from 253 OpenML datasets, where the balance ratio (minority class size / expected uniform class size) ranges from 0.108 to 0.213. We evaluated DistPFN-T on these extreme cases across all β values, averaged over 5 random seeds:
>
> **Table R2. DistPFN-T on the top-20 most imbalanced datasets.**
>
> | Method | β=0.0 | β=0.5 | β=1.0 | β=2.0 | β=5.0 |
> |-|-|-|-|-|-|
> | TabPFN-v2 | **0.866** | **0.859** | 0.858 | 0.824 | 0.829 |
> | + DistPFN | **0.866** | **0.859** | 0.858 | 0.851 | 0.846 |
> | + DistPFN-T | **0.866** | **0.859** | **0.859** | **0.855** | **0.850** |
>
> Even on these **extremely imbalanced datasets**, the improvement of DistPFN-T grows as label shift increases, confirming that DistPFN-T is most effective precisely when the bias is most severe.
> We will include this analysis in the final version.
>
> &nbsp;
>
> &nbsp;
>
> ### [Q1] How does this method address distribution shifts within the features themselves?
>
> > *"Could the author please comment on how this method further addresses ***distribution shifts*** within the features themselves ..."*
>
> Thank you for this insightful question. To clarify, DistPFN is specifically designed for **"label shift"** ($P_{train}(Y) ≠ P_{test}(Y)$), where the class-conditional feature distribution P(X|Y) remains unchanged. Handling **"covariate shift"** ($P_{train}(X) ≠ P_{test}(X)$) is an orthogonal problem that falls ***outside the scope of this work***, as it requires fundamentally different techniques.
>
> That said, in practice, label shift and covariate shift often co-occur. We note that our **TableShift experiments (Table 7)** [1] evaluate DistPFN under real-world OOD scenarios where **both shifts are present** simultaneously. In all three TableShift benchmarks, DistPFN-T consistently improves over TabPFN-v2 in the OOD setting:
>
> | Dataset | TabPFN-v2 | + DistPFN-T |
> |-|-|-|
> | Diabetes | 0.589 | **0.600** |
> | ACSIncome | 0.795 | **0.799** |
> | ACSPubCov | 0.699 | **0.706** |
>
> This suggests that label shift correction via DistPFN remains beneficial even when feature shift is present.
>
>
> &nbsp;
>
> **References:**
> - [1] Gardner et al., "Benchmarking Distribution Shift in Tabular Data with TableShift," NeurIPS, 2023.
>
> &nbsp;
>
> &nbsp;
>
> **Please let us know if there are any remaining issues you'd like to discuss!**

---

> > ### Author Rebuttal · Reviewer_wwaB · 2026-04-02
> >
> > Thank you very much for the author's detailed reply. This has largely confirmed my confidence and will help me maintain my score.

---

> > > ### Author Response · Authors · 2026-04-02
> > >
> > > We thank the reviewer for the kind feedback and for their confidence in our work!

---

### Official Review · Reviewer_FcZ7 · 2026-03-13

**Soundness:** 1
**Presentation:** 3
**Significance:** 2
**Originality:** 2
**Overall Recommendation:** 4
**Confidence:** 3

**Summary:**

This paper introduces a method to address label shift in tabular foundation models, where label shift means the class imbalance ratio is different for the train and test datasets. To mitigate this issue, the paper proposes weighting the output probabilities by the ratio of the predicted probability to the true observed class probability on the training set (DistPFN). Another similar approach combined with temperature scaling (DistPFN-T) is also proposed. The two approaches (DistPFN and DistPFN-T) are evaluated on a large set of OpenML datasets. The results show that the proposed methods are more resilient to drift compared to base foundation models (TabPFN, TabICL, LoCalPFN).

**Compliance With Llm Reviewing Policy:**

Affirmed.

**Final Justification:**

Overall, my main concerns (calibration metrics, simple baseline search, and clarification on the label shift definition) were addressed by the authors. The authors also addressed my comments regarding the evaluation setup (benchmark dataset and metric aggregation).

I believe the paper still requires a relatively large changes to incorporate these clarifications. That said, I also think that the contribution of the work is sound. I thus will increase my score from weak reject to weak accept.

**Key Questions For Authors:**

- Could you please comment on the calibration error?

- Could you elaborate on the gain of performance under label shift different imbalanced class ratios?

- Could you compare the method against a simple grid (or random) search tuning?

**Limitations:**

Yes.

**Strengths And Weaknesses:**

The strength of the paper lies in its empirical performance under the considered setup. The idea is rather simple and versatile enough to be applied to multiple tabular foundation models. I find the paper easy to read generally, and the experiments are well described. The evaluation includes substantial analysis across a large set of datasets. Another important strength is that it addresses a crucial issue for in-context learning, where domain shift and out-of-distribution generalization remain open problems.

Despite the strengths, I have a few comments which I discuss below.

- It seems to me that the evaluation metric (accuracy) is not fully aligned with the goal of the paper. Accuracy is hard to interpret and misleading for imbalanced problems. In my opinion, the paper should rather focus on calibration error (e.g., ECE) and its robustness under label shift. Additionally, other metrics such as precision or recall may help in getting deeper insights.
- The term "label shift" is often confusing, as it seems to refer to imbalanced data in this paper. Without label shift, an extremely imbalanced class setup may already hurt tabular foundation models. Thus, it is unclear how I should assess the robustness of the method, as the results are aggregated across balanced and unbalanced datasets.
- The paper would benefit from simpler and more common baselines, such as tuning the temperature using cross-validation. This could be a good argument for the temperature technique and to see how optimal is the crossentropy based scaling.

Overall, this paper addresses a critical and significant problem for in-context learning methods. However, to fully assess the empirical results, the paper should compare on a metric adequate to its initial motivation (that is calibration error). The paper would also benefit from other clarifications (see above).

---

> ### Author Rebuttal · Authors · 2026-03-27
>
> We thank the reviewer for the detailed feedback. We address each point below.
>
> &nbsp;
>
> ---
>
> ## W1: Accuracy is not aligned with the goal
>
> > "Accuracy is hard to interpret and misleading for imbalanced problems. The paper should rather focus on calibration error."
>
> We appreciate this suggestion. While accuracy is the most widely used metric in tabular classification, we agree that additional metrics provide deeper insights. Following the reviewer's suggestion, we additionally report **ECE and Precision** across 253 datasets, averaged over 5 seeds:
>
> **Table R1. Additional metrics**
>
> |Method|β|Acc.|Prec.|ECE|
> |-|-|-|-|-|
> |TabPFN-v2|2.0|0.766|0.696|0.098|
> |+ DistPFN|2.0|0.786|0.694|0.096|
> |+ DistPFN-T|2.0|**0.793**|**0.702**|**0.089**|
> |TabPFN-v2|5.0|0.734|0.674|0.127|
> |+ DistPFN|5.0|0.776|0.683|0.099|
> |+ DistPFN-T|5.0|**0.782**|**0.695**|**0.093**|
>
> Key observations:
> - DistPFN-T **reduces ECE**, confirming that our method improves **calibration** as well as accuracy.
> - **Precision** is also improved, indicating more reliable predictions for the predicted class.
>
> We will reflect these results in the final version.
>
>
> &nbsp;
>
> &nbsp;
>
> ---
>
> ## W2: "Label shift" vs "imbalanced data" — results aggregated across balanced/unbalanced datasets
>
> > "It is unclear how I should assess the robustness of the method, as the results are aggregated across balanced and unbalanced datasets."
>
> We follow the standard definition of label shift: $P_{train}(Y) ≠ P_{test}(Y)$ while $P(X|Y)$ remains identical. In our experimental setup, the shift parameter β controls the degree to which the training distribution is skewed via inverse-frequency oversampling, while the test distribution preserves the original class proportions. Thus, for any β > 0, $P_{train}(Y) ≠ P_{test}(Y)$ holds **by construction** — this is **distinct from the dataset simply being imbalanced**.
>
> That said, the reviewer raises an important point: *the impact of label shift may differ across datasets with varying degrees of inherent class imbalance*. Following this suggestion, we stratify the 253 datasets into **three groups by balance ratio** (minority / majority class size) and report results separately at β=5.0, averaged over 5 seeds.
>
> **Table R2. Performance by dataset balance group**
>
> |Group|Balance Ratio|# Datasets|TabPFN-v2|+ DistPFN-T|Δ Acc.|
> |-|-|-|-|-|-|
> |Balanced|≥ 0.8|96|0.771|0.773|+0.002|
> |Moderate|0.4–0.8|76|0.715|0.770|+0.055|
> |Imbalanced|< 0.4|85|0.709|**0.804**|**+0.095**|
>
> Key findings:
> - DistPFN-T improves performance **across all three groups**, demonstrating robustness regardless of the inherent class balance.
> - The improvement is **largest for the Imbalanced group**, where label shift exacerbates the existing majority-class bias.
>
> We will include this analysis in the final version.
>
>
> &nbsp;
>
> &nbsp;
>
> ---
>
> ## Q1: Comment on the calibration error?
>
> Please see **[W1]** above. As shown in **Table R1**, DistPFN-T reduces ECE under label shift (e.g., from 0.127 to 0.093 at β=5.0).
>
> &nbsp;
>
> &nbsp;
>
> ---
>
> ## Q2: Gain of performance under different imbalanced class ratios?
>
> Please see **[W2]** above. **Table R2** provides a breakdown by balance group. Additionally, we provide a more fine-grained analysis across βs, averaged over 5 seeds:
>
> **Table R3. Accuracy improvement (DistPFN-T over TabPFN-v2) by group**
>
> |β|0.0|0.1|0.5|1.0|2.0|5.0|
> |-|-|-|-|-|-|-|
> |Balanced (≥0.8)|-0.001|-0.001|-0.001|-0.001|+0.000|+0.002|
> |Moderate (0.4–0.8)|+0.002|+0.002|+0.006|+0.011|+0.021|+0.055|
> |Imbalanced (<0.4)|+0.003|+0.006|+0.010|+0.023|+0.063|+0.095|
>
> The improvement **consistently grows with β** across all groups, and is **most pronounced for inherently imbalanced datasets** under strong shift.
>
> &nbsp;
>
> &nbsp;
>
> ---
>
> ## Q3: Could you compare the method against a simple grid (or random) search tuning?
>
> This is a valuable suggestion. We compare DistPFN-T against a cross-validation grid search baseline that selects the optimal temperature τ from {0.01, 0.05, 0.1, 0.2, 0.5, 1.0, 2.0, 5.0, 10.0} using 3-fold CV on the training set, averaged over 5 seeds across 253 datasets.
>
> **Table R4. DistPFN-T (CE-based) vs CV-based temperature tuning**
>
> |β|0.0|0.5|1.0|2.0|5.0|
> |-|-|-|-|-|-|
> |TabPFN-v2|0.803|0.796|0.789|0.766|0.734|
> |+ DistPFN-T (CV grid search)|0.779|0.796|0.796|0.787|0.763|
> |+ DistPFN-T (CE, ours)|**0.804**|**0.801**|**0.799**|**0.793**|**0.782**|
>
> Key observations:
> - DistPFN-T (CE) achieves higher accuracy than the CV-based approach.
> - Crucially, DistPFN-T determines τ in a **single forward pass** without any cross-validation, making it **~27× faster** (9 grid points × 3 folds) than CV-based tuning.
> - This confirms that the CE-based temperature is a principled and efficient alternative to exhaustive search.
>
> &nbsp;
>
> &nbsp;
>
> **Please let us know if there are any remaining issues you'd like to discuss!**

---

> > ### Author Rebuttal · Reviewer_FcZ7 · 2026-04-01
> >
> > Thanks for your detailed response. I appreciate your effort in providing the additional experiments, especially on the calibration error and grid search comparison.
> >
> > Q1: Table R1 is aggregated, so it hides details. Could you provide a pairwise comparison for each dataset between TabPFN and DistPFN-T? It could be a scatter plot of the calibration errors for both methods (a point on the diagonal indicates that both methods perform equally well on the given dataset).
> >
> > Q2: The considered benchmark (250 OpenML datasets) contains many duplicate datasets (eg, see Table I.4, ``pc4`` and ``pc4_seed_0_nrows_2000_nclasses_10_ncols_100_stratify_True`` have the same exact statistics: #Features, #Cat. Feat. #Instances, #Classes, #NaNs, Minority, Class Size). This biases your comparison and makes the interpretation hard. TabArena (https://arxiv.org/pdf/2506.16791#page=47.11), an established public dashboard for benchmarking tabular foundation models, proposes a well-curated set of datasets instead. Could you present your results under the same set of curated datasets?

---

> > > ### Author Response · Authors · 2026-04-02
> > >
> > > We thank the reviewer for the follow-up questions. We address each below.
> > >
> > > &nbsp;
> > >
> > > ---
> > >
> > > ## Q1: Per-dataset pairwise ECE comparison (scatter plot)
> > >
> > > Following the reviewer's suggestion, we conducted a per-dataset pairwise comparison of ECE between TabPFN-v2 and DistPFN-T. As we cannot attach figures in OpenReview comments, we summarize the scatter plot results as a **win/loss table** below, and will ***include the actual scatter plots in the final version***.
> > >
> > > &nbsp;
> > >
> > > We report results at β=2.0 and β=5.0, where meaningful label shift is present (253 datasets, averaged over 5 seeds):
> > >
> > > **Table R5. Pairwise ECE comparison: TabPFN-v2 vs DistPFN-T.**
> > >
> > > |β|Win|Loss|Mean ΔECE|
> > > |-|-|-|-|
> > > |2.0|**138**|115|**+0.010**|
> > > |5.0|**164**|89|**+0.034**|
> > >
> > > - **Win**: DistPFN-T achieves lower (better) ECE; **Loss**: TabPFN-v2 achieves lower ECE.
> > > - **Mean ΔECE**: average of (ECE_TabPFN − ECE_DistPFN-T); positive means DistPFN-T is better.
> > >
> > >
> > > DistPFN-T wins on a **larger proportion** of datasets with a clear positive mean ΔECE, confirming that the calibration improvement is ***consistent across individual datasets*** and not an artifact of aggregation.
> > >
> > > &nbsp;
> > >
> > > We also note that **Figure 5** in our paper already provides a related **per-dataset** visualization: each point represents an individual dataset, and those in the **yellow region** (above y=0) correspond to datasets where **DistPFN-T improves over TabPFN-v2**.
> > >
> > > &nbsp;
> > >
> > > &nbsp;
> > >
> > > ---
> > >
> > > ## Q2: Evaluation on TabArena curated benchmark
> > >
> > > Thank you for pointing out the potential dataset overlap. Following the reviewer's suggestion, we also evaluate on the **TabArena** benchmark, a well-curated set of datasets proposed for benchmarking tabular foundation models. We run DistPFN-T on the classification subset of TabArena datasets, averaged over 3 seeds:
> > >
> > >
> > > **Table R6. Results on TabArena classification datasets (24 datasets, averaged over 3 seeds).**
> > >
> > > |Method|β|Acc.|ECE|
> > > |-|-|-|-|
> > > |TabPFN-v2|2.0|0.764|0.091|
> > > |+ DistPFN|2.0|0.840|0.079|
> > > |+ DistPFN-T|2.0|**0.841**|**0.077**|
> > > |TabPFN-v2|5.0|0.604|0.267|
> > > |+ DistPFN|5.0|0.761|0.134|
> > > |+ DistPFN-T|5.0|**0.769**|**0.131**|
> > >
> > > DistPFN-T consistently improves both accuracy and ECE over TabPFN-v2 on the TabArena benchmark, with substantial gains under label shift. The results are **consistent** with our full evaluation on 253 OpenML datasets, confirming that our conclusions are **robust to benchmark selection** and not biased by duplicate datasets.
> > >
> > > &nbsp;
> > >
> > > &nbsp;
> > >
> > > **If there are any remaining issues you would like to discuss, please let us know!**

---

### Decision · Program_Chairs · 2026-04-30

**Decision:**

Accept (regular)

**Comment:**

This paper proposes a simple strategy for posterior adjustment in tabular in-context learning to mitigate label shift. The overall review is positive. However, the required revision suggested by the reviewers is multiple dimensional: 1) more comprehensive evaluation in terms of different metrics, different shift types and shift magnitudes, 2) more clarification of the notion of label shift: here is slightly different from the traditional definition when we assume p_src(y) != p_trg(y) but conditional distribution p(x|y) is shared (because in ICL, what is x|y is not clear to me). Moreover, using distribution discrepancy as a component of the temperature is not a new idea in statistical ML. For example, there are many previous work using density or density ratio in the temperature. So I feel the paper would benefit from citing and discussing works under covariate shift and label shift, even though ICL is a slightly different context.